# Toward Exploratory Inverse Constraint Inference with Generative Diffusion Verifiers

**Runyi Zhao**[1*]**, Sheng Xu**[1*]**, Bo Yue**[1]**, Guiliang Liu**[1†]
[1]School of Data Science, The Chinese University of Hong Kong, Shenzhen
`{runyizhao,shengxu1,boyue}@link.cuhk.edu.cn`
`liuguiliang@cuhk.edu.cn`

## Abstract

An important prerequisite for safe control is aligning the policy with the underlying constraints in the environment. In many real-world applications, due to the difficulty of manually specifying these constraints, existing works have proposed recovering constraints from expert demonstrations by solving the Inverse Constraint Learning (ICL) problem. However, ICL is inherently ill-posed, as multiple constraints can equivalently explain the experts' preferences, making the optimal solutions not uniquely identifiable. In this work, instead of focusing solely on a single constraint, we propose the novel approach of Exploratory ICL (ExICL). The goal of ExICL is to recover a diverse set of feasible constraints, thereby providing practitioners the flexibility to select the most appropriate constraint based on the practical needs of deployment. To achieve this goal, we design a generative diffusion verifier that guides the trajectory generation process using the probabilistic representation of an optimal constrained policy. By comparing these decisions with those made by expert agents, we can efficiently verify a candidate constraint. Driven by the verification feedback, ExICL implements an exploratory constraint update mechanism that strategically facilitates diversity within the collection of feasible constraints. Our empirical results demonstrate that ExICL can seamlessly and reliably generalize across different tasks and environments. The code is available at `https://github.com/ZhaoRunyi/ExICL`.

## 1 Introduction

In recent years, Reinforcement Learning (RL) agents have demonstrated remarkable performance in a variety of virtual games and environments by extensively exploring and exploiting the entire state-action space (Mnih et al., 2015; Silver et al., 2018; Vinyals et al., 2019). However, real-world applications often prioritize the safety and reliability of decisions, requiring RL policies to operate under restricted regions or spaces in realistic environments. To learn such policies, safe RL methods typically update the policy within the bounds of constraints (Liu et al., 2021). However, in practical applications, these constraints are often not readily available and can be challenging to specify manually, particularly in complex environments.

Recent advances in Inverse Constraint Learning (ICL) propose recovering the constraints followed by expert agents from their demonstration (Scobee & Sastry, 2020). Previous methods (Malik et al., 2021; Kim et al., 2023) typically extend the classical Inverse Reinforcement Learning (IRL) framework to learn the constraint model under known rewards. However, IRL is essentially an ill-posed problem (Ng & Russell, 2000), and the optimal solution is often non-identifiable. When it comes to ICL, we find multiple constraints can equivalently explain expert demonstrations, which makes it difficult to identify the real constraints. In resolving the problem of unidentifiable solution, previous ICL solvers typically rely on additional assumptions, such as the expert agent implementing a regularized policy (Malik et al., 2021), or the ground-truth constraint set having a minimum coverage of state-action pairs (Scobee & Sastry, 2020). While these assumptions reduce the number of candidate constraints, there is no guarantee that the real constraint can be uniquely identified or accurately characterized by these assumptions.

---

*Equal contributions. †Corresponding author: Guiliang Liu, liuguiliang@cuhk.edu.cn.

To address these issues, in this work, we propose an Exploratory Inverse Constraint Learning (Ex-ICL) algorithm, which learns the set of feasible constraints with which the agent can accurately recover the expert demonstration. To deploy these constraints in practical applications, practitioners can select from the feasible set based on domain knowledge or specific requirements. Although the concept of a feasible solution set has been theoretically analyzed by IRL solvers (Metelli et al., 2021) and extended to ICL settings (Yue et al., 2024; 2025), the development of practical implementations remains largely unexplored. This is due to two main reasons (see Figure 1): 1) **The difficulty in verification**: To verify the feasibility of a candidate constraint, prior studies must tackle a forward constrained RL problem. This involves multiple rounds of policy model updates under the constraint, rendering the recovery of a feasible set of constraints computationally intractable, especially in complex environments. 2) **The lack of exploratory mechanisms**: Previous ICL algorithms focused primarily on identifying a single constraint, lacking an exploratory mechanism to infer a diverse set of feasible constraints.

In response to the difficulty in verification, in this work, our ExICL algorithm utilizes a Generative Diffusion Verifier (GDV) to accelerate verification. Operating within an in-context learning framework, GDV can generate the optimal trajectory for a given constraint model without necessitating updates to model parameters. This capability is enabled by a generative optimization framework, where the GDV guides the generative process of a diffusion model with the probabilistic representation of an optimal policy under a specified constraint. During the trajectory generation process, the constraint model must accurately predict the cost of trajectories under varying levels of noise. Motivated by this requirement, we designed a noise-robust objective for updating constraints. The divergence between the generated trajectories and the expert trajectories efficiently determines the feasibility of a constraint and guides the updates to our constraint model.

Upon identifying a feasible constraint, our ExICL algorithm initiates an exploratory update to identify additional constraints. To facilitate this, we develop a strategic exploration mechanism that enhances the diversity of feasible constraints learned through a contrastive learning objective. In particular, we characterize each constraint by the predicted feasibility (or cost) values assigned to each state-action pair. Building on this characterization, we refine the constraint update objective to encourage divergence between these values and those of other constraints within the feasible set. This exploratory update mechanism enables learning of a diverse set of feasible constraints.

To empirically validate our ExICL method, we assess its performance across a diverse set of tasks (including navigation, locomotion, and autonomous driving) and under various types of constraints (such as spatial, dynamic, and kinematic). The results demonstrate that ExICL can outperform other baseline methods from multiple perspectives, including 1) inferring more accurate constraints, 2) enhancing the diversity of the inferred constraint set, and 3) boosting training efficiency. These empirical studies consolidate the validity of our ExICL.

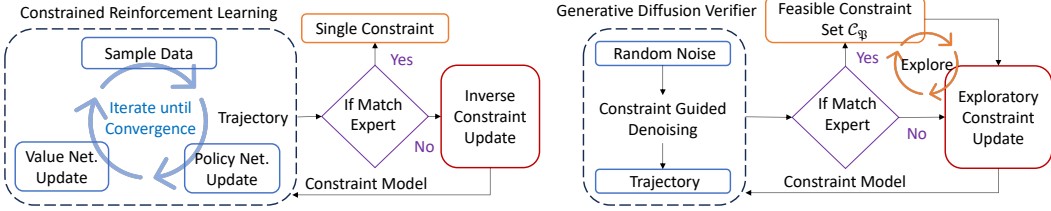

Figure 1: The flowchart of classic ICL (left) and our ExICL (right). The key differences are: 1) To verify a constraint, ICL iteratively updates the policy and value functions using samples from the dataset, whereas ExICL employs a guided generation approach, simplifying the verification process. 2) while the objective of classic ICL is to infer a single constraint model, ExICL adopts an exploratory constraint learning strategy aimed at identifying a broader, feasible set of constraints.

## 2 PROBLEM FORMULATION

**Constrained Reinforcement Learning (CRL).** The CRL problem commonly studies a Constrained Markov Decision Process (CMDP) setting $\mathcal{M}_c := (\mathcal{S}, \mathcal{A}, \mathcal{T}, r, c, \epsilon, \mu_0, \gamma)$, where: 1) $\mathcal{S}$ and $\mathcal{A}$ denote the space of states and actions. 2) $\mathcal{T}(s'|s, a)$ defines the transition distribution. 3) $r(s, a)$ and

$c(s, a)$ denote the reward and cost function (we assume $c \geq 0$). 4) $\epsilon$ defines the threshold of the constraint, where $\epsilon = 0$ refers to a hard constraint, enforcing absolute satisfaction, while $\epsilon > 0$ denotes a soft constraint, permitting a certain degree of constraint violation. 5) $\mu_0$ denotes the initial state distribution. 6) $\gamma \in [0, 1)$ is the discount factor. In our empirical study, we mainly study an episodic setting where the game ends at some terminating state or time horizon. The goal of CRL policy $\pi \in \Delta_{\mathcal{S}}^{A}$ is to maximize the expected discounted rewards under known constraints

$$\max_{\pi} \mathbb{E}_{\mathcal{T}, \pi, \rho_0} \left[ \sum_{t=0}^{T} \gamma^t r(s_t, a_t) \right] \text{ s.t. } \mathbb{E}_{\mathcal{T}, \pi, \rho_0} \left[ \sum_{t=0}^{T} \gamma^t c(s_t, a_t) \right] \leq \epsilon \qquad \text{(PI)}$$

**Inverse Constraint Learning.** While traditional CRL problems often assume that cost signals $c(\cdot)$ are directly observable from the environment, in many real-world scenarios, we typically have access to expert demonstrations $\mathcal{D}_E$ rather than observing the cost signals directly. To address this challenge, a recent study (Scobee & Sastry, 2020) introduced Inverse Constrained Reinforcement Learning (ICRL). The objective of ICRL is to infer the underlying constraint models from expert demonstrations, enabling any policy to reproduce these demonstrations by applying the recovered constraints. To achieve it, previous works (Malik et al., 2021) apply the Maximum likelihood Estimation (MLE) objective under the maximum entropy framework:

$$\arg \max_{c} p_{\mathcal{M}_c}(\mathcal{D}_E | C) = \arg \max_{c} \frac{1}{Z_c^{|\mathcal{D}_E|}} \prod_{\tau_E \in \mathcal{D}_E} e^{r(\tau_E)} \mathbb{1}^{\mathcal{M}_c}(\tau_E) \qquad (1)$$

Inspired by (Malik et al., 2021), we approximate $\mathbb{1}^{\mathcal{M}_c}(\tau) = \prod_{t=0}^{T} \phi_\omega(s_t, a_t)$ such that $\phi_\omega(s_t, a_t) \in [0, 1]$ indicates the permissibility of performing action $a_t$ at a state $s$. The MLE gradient $\nabla_\omega p_{\mathcal{M}_c}(\mathcal{D}_E | C)$ can be transformed to:

$$\nabla_\omega p_{\mathcal{M}_c}(\mathcal{D}_E | C) = \mathbb{E}_{\tau^* \sim \mathcal{D}^E} \left[ \nabla_\omega \log \phi_\omega(\tau^*) \right] - \mathbb{E}_{\hat{\tau} \sim (\pi_{\mathcal{M}_c}, \mathcal{T})} \left[ \nabla_\omega \log \phi_\omega(\hat{\tau}) \right] \qquad (2)$$

where 1) the cost function can derived by $c(s, a) = -\log \phi(s, a)$, 2) $\hat{\tau}$ denotes the estimated trajectory based on the CMDP $\mathcal{M}_c$ with the learned cost $c$. Under this setting, ICRL algorithms typically assume that *reward signals are observable and the goal is to recover only the constraints*, in contrast to Inverse Reinforcement Learning (IRL) (Ziebart et al., 2008), which aims to learn rewards from an unconstrained MDP.

**Identifiability Issue in ICRL.** Like many other inverse optimization problems (Arora & Doshi, 2021), ICRL is essentially ill-posed since various combinations of rewards and constraints can explain the same expert demonstrations, which makes it difficult to identify the ground-truth constraint uniquely. Striving for the identifiability of solutions, a pioneer work (Scobee & Sastry, 2020) introduced the concept of a minimum constraint under a discrete CMDP. This constraint comprises the smallest number of state-action pairs necessary for an expert agent to reproduce expert demonstrations, ensuring that the optimal constraint is as concise as possible. A continuing work (Malik et al., 2021) extends the concept of the minimum constraint to continuous state-action spaces by incorporating a weighted regularizer into the constraint update objective. However, this extension faces several challenges: 1) Finding the minimum constraint in a continuous state space is intractable. 2) The scale of regularization is highly sensitive to the chosen weighting term, which can not guarantee the solution is unique. 3) Even when we capture the minimum constraint, there is no guarantee that this constraint is the one respected by the actual expert agents.

Beyond focusing on the exact minimum constraints, inspired by the recent theoretical advancement in IRL (Metelli et al., 2021), an intriguing but less explored solution is to infer the set of feasible constraints. Under this setting, recent studies (Lindner et al., 2022; Metelli et al., 2023) developed a theoretical framework for characterizing the feasible set of solutions for inverse optimization problems. However, these studies focus on reward learning instead of the constraint inference problem. Besides, their results are only applicable under discrete state-action spaces, but the realistic application aligns better with continuous state-action space. In the meantime, while previous works commonly rely on online interaction with the environment, we focus on an offline setting. This is because in practice data collection is expensive (e.g., in robotics, educational agents, or healthcare) or dangerous (e.g., in autonomous driving, or healthcare). Motivated by the above considerations, we propose the following offline ICRL problem considering the feasible constraint set.

**Definition 2.1.** The problem of *offline inference for feasible constraint set* can be characterized by a pair $\mathfrak{P} = (\mathcal{M}, \mathcal{D}^0)$, where $\mathcal{M}$ is a CMDP\$c$ (CMDP without the cost) and $\mathcal{D}^0 = \{\mathcal{D}^E, \mathcal{D}^{-E}\}$ is the

offline demonstration such that $\mathcal{D}^E = \{s_n^E, a_n^E, r_n^E\}_{n=1}^{N_E}$ denotes expert demonstrations and $\mathcal{D}^{-E}$ denotes the dataset generated by non-expert agents. A cost model $c : \mathcal{S} \times \mathcal{A} \times H \rightarrow \mathbb{R}$ is feasible for $\mathfrak{P}$ if $\pi^E$ is an optimal policy for the CMDP $\mathcal{M} \cup c$, i.e., $\pi^E \in \Pi_{\mathcal{M} \cup c}^*$. We denote by $\mathcal{C}_{\mathfrak{P}}$ the set of feasible cost functions for $\mathfrak{P}$, namely feasible constraint set.

In solving the ICSI problem above, a critical prerequisite is to efficiently assess the feasibility of a constraint by determining whether an expert policy can be learned under this constraint. However, traditional ICRL solvers (Scobee & Sastry, 2020; Malik et al., 2021; Liu et al., 2024) are based on a bi-level optimization framework, which requires solving both a forward CRL problem and an Inverse Constraint Learning problem. Updating both a policy and a constraint function is computationally intensive, significantly impacting the efficiency of learning the feasible constraint set.

## 3 EXPLORATORY INVERSE CONSTRAINT LEARNING

To learn a diverse set of feasible constraint, we introduce the Exploratory Inverse Constraint Learning (ExICL) algorithm (1). Striving for efficient assessment of candidate constraints, ExICL leverages a Generative Diffusion Verifier (GDV) to evaluate whether expert demonstrations can be reproduced under the examined constraint (Section 3.1). This optimization operates in context, thus it bypasses the need to update model parameters, resulting in a significant increase in computational efficiency. To ensure that the constraint model can effectively guide the denoising process, ExICL implements a noise-robust constraint learning (Section 3.2). Additionally, ExICL employs an iterative exploration process that strategically updates the constraint function, thereby facilitating the identification of a broad range of feasible and diverse constraints (Section 3.3).

### 3.1 DIFFUSION PLANNER FOR GENERATIVE VERIFICATION

**Learning the Planner from Offline Demonstration.** Inspired by the Diffuser (Janner et al., 2022; Ho et al., 2020), we follow the diffusion probabilistic models (Sohl-Dickstein et al., 2015; Ho et al., 2020) and formulate planning as a trajectory generation task through a learned iterative denoising diffusion process $p_\theta(\tau^{i-1}|\tau^i)$. The data distribution induced by the denoising process is given by:

$$p_\theta(\tau^{0:I}) = p_\theta(\tau^I) \prod_{i=1}^{I} p_\theta(\tau^{i-1}|\tau^i) \quad \text{where} \quad p_\theta(\tau^{i-1}|\tau^i) = \mathcal{N}(\mu_\theta(\tau^i, i), \Sigma^i) \tag{3}$$

where $i \in [0, I]$ denotes the diffusion step and each $\tau^i = (s_0^i, a_0^i, s_1^i, a_1^i, \ldots, s_T^i, a_T^i)$ where $t \in [0, T]$ denotes the planning step (i.e., the time step of action execution in the environment).

This learned denoising process is trained to reverse a forward diffusion process $q(\tau^i|\tau^{i-1})$ that slowly corrupts the structure of trajectories by adding noise. The corresponding distribution is:

$$q(\tau^{0:I}) = q(\tau^0) \prod_{i=0}^{I-1} q(\tau^i|\tau^{i-1}) \quad \text{where} \quad q(\tau^i|\tau^{i-1}) = \mathcal{N}(\sqrt{1-\beta_i}\tau^{i-1}, \beta_i I) \tag{4}$$

Under these denoising and diffusion processes, $q(\tau^0)$ denotes the data distribution, and $q(\tau^I)$ indicates the standard Gaussian prior. To learn the denoising model $p_\theta(\tau^{i-1}|\tau^i)$ from the offline dataset $\mathcal{D}^0$, we optimize its parameters by maximizing the negative log-likelihood of observed trajectories via constructing a variational lower bound over the individual steps of denoising:

$$\mathbb{E}_{q(\tau^0)}[\log p(\tau^0)] \geq \mathbb{E}_{q(\tau^{0:I})}\left[\log \frac{p_\theta(\tau^{0:I})}{q(\tau^{1:I}|\tau^0)}\right] \approx \mathbb{E}_{\tau^0 \sim \mathcal{D}^0, \tau^{1:N} \sim q(\tau^{1:N})}\left[\sum_{i=1}^{I} \log p_\theta(\tau^{i-1}|\tau^i)\right] \tag{5}$$

Note that the $p_\theta(\tau) = \int p_\theta(\tau^{0:I})\tau^{1:I}$ approximate the distribution of trajectories within the offline demonstration $\mathcal{D}^0$, and there is no guarantee a trajectory $\tau \sim p_\theta(\tau)$ generated under this model will be safe. By leveraging this property, GDV can efficiently verify the feasibility of a constraint by determining whether it can guide the diffusion planning process to generate a safe trajectory.

**Verifying Constraint via Guided Denoising.** To verify the feasibility of a constraint, GDV utilizes the guided sampling strategy (Janner et al., 2022) and perturbs the distributions in the iterative denoising process. At each denoising step, the perturb distribution $\tilde{p}_\theta(\tau)$ can be represented as:

$$\tilde{p}_\theta(\tau) = p(\tau|\mathcal{O}_{0:T} = 1) \propto p_{\mathcal{M}_c}(\mathcal{O}_{0:T} = 1|\tau)p_\theta(\tau) \tag{6}$$

where $\mathcal{O}_{0:T}$ is a binary variable denoting whether the outcomes are desired. Without considering the constraint, the probabilistic inference framework for RL (Levine, 2018) defines

$$p(\mathcal{O}_{0:T} = 1, \tau) = p_\theta(\tau) p(\mathcal{O}_{0:T} = 1 | \tau) = p(s_0) \left[ \prod_{t=0}^{T} p(s_{t+1}|s_t, a_t) \pi_\beta(a_t|s_t) \right] e^{\sum_{t=0}^{T} r(s_t, a_t)} \quad (7)$$

where 1) $\pi_\beta(a_t|s_t)$ denotes the behavior policy that generates the offline dataset. 2) $p_\theta(\tau) = p(s_0) \left[ \prod_{t=0}^{T} p(s_{t+1}|s_t, a_t) \pi_\beta(a_t|s_t) \right]$ denotes the trajectory distribution in the offline dataset $\mathcal{D}^0$, and 3) $p(\mathcal{O}_{0:T} = 1 | \tau) = e^{\sum_{t=0}^{T} r(s_t, a_t)}$ the optimality model.

By extending this advancement to a CMDP $\mathcal{M}_c$, the optimal probabilistic representation for constrained policy model $p_{\mathcal{M}_c}(\mathcal{O}_{0:T} = 1 | \tau)$ is defined by:

$$p_{\mathcal{M}_c}(\mathcal{O}_{0:T} = 1 | \tau) = \begin{cases} e^{\sum_{t=0}^{T} r(s_t, a_t)}, & \mathbb{E}_\tau[\sum_{t=0}^{T} c(s_t, a_t)] \leq \epsilon \\ 0, & \mathbb{E}_\tau[\sum_{t=0}^{T} c(s_t, a_t)] > \epsilon. \end{cases} \quad (8)$$

Although the CRL objective is essentially non-convex, this problem in general has zero duality gap:

**Theorem 3.1.** *(CRL has zero duality gap (Paternain et al., 2019)). Suppose that $r$ and $c$ are bounded and the Slater's condition holds for (PI), then strong duality holds for (PI), i.e., PI\* = DI\*.*

$$\min_{\lambda > 0} \max_{\pi} \mathbb{E}_{\mu_0, \pi, P_\mathcal{T}} \left[ \sum_{t=0}^{h} \gamma^t \Big( r(s_t, a_t) - \lambda c(s_t, a_t) \Big) \right] + \lambda \epsilon \quad \text{(DI)}$$

Correspondingly, $p_{\mathcal{M}_c}(\mathcal{O}_{0:T} = 1 | \tau)$ (Equation 8) can be represented as its dual format by transforming the constraint into penalty such that:

$$\tilde{p}_{\mathcal{M}_c}(\tau) \propto p_\theta(\tau) p_{\mathcal{M}_c}(\mathcal{O}_{0:T} = 1 | \tau) = p_\theta(\tau) e^{\sum_{t=0}^{T} [r(s_t, a_t) - \lambda c(s_t, a_t) + \lambda \epsilon]} \quad (9)$$

where $\lambda$ denotes the Lagrange multiplier. $\tilde{p}_{\mathcal{M}_c}(\tau)$ represents the distribution of trajectories subjected to reward maximizing and constraint satisfying objectives, which can model the trajectories generated under a given constraint. Since $\tilde{p}_{\mathcal{M}_c}(\tau)$ is modeled by a diffusion model, the denoising process transitions can be approximated as Gaussian (Sohl-Dickstein et al., 2015) such that:

$$p_\theta(\tau^{i-1} | \tau^i, \mathcal{O}_{0:T}) = \mathcal{N}(\mu_\theta(\tau^i, i) + \Sigma g_c, \Sigma^i) \text{ where } g_c = \nabla_\tau p_{\mathcal{M}_c}(\mathcal{O}_{0:T} = 1 | \tau) \mid_{\tau = \mu_\theta} \quad (10)$$

Such a denoising process does not involve parameter updating over the diffusion or cost/reward-value models, and thus it is more efficient than the classic CRL solver. By comparing the generated trajectories $\tilde{\tau} \sim \tilde{p}_{\mathcal{M}_c}(\tau)$ with the expert ones $\tau^E$, GDV efficiently validates the accuracy of inferred constraints. During this process, to ensure that the GDV can accurately guide trajectory generation and facilitate the discovery of a diverse set of feasible constraints, we must ensure the following:

- The constraint model can accurately guide the trajectory generation in the GDV. Specifically, the model must be able to predict the cost of trajectories with added Gaussian noise accurately.
- Our algorithm includes a mechanism to generate a large number of candidate constraints.

In the following sections, we introduce the approach to learning noise-robust constraints with $\tilde{p}_{\mathcal{M}_c}(\tau)$ and the exploration strategy for learning a diverse set of constraints.

## 3.2 NOISE-ROBUST CONSTRAINT UPDATE

To ensure that the constraint model accurately estimates the cost of noisy trajectory, we collect noisy samples ($\tau^{*, 0:I}$ and $\hat{\tau}^{0:I}$) during the diffusion process by introducing noise into $\tau^*$ and $\hat{\tau}$. Subsequently, we update the classic constraint inference loss (2) to a noise-robust version in the following:

$$\mathbb{E}_{\tau^* \sim \mathcal{D}^E} \left[ \mathbb{E}_{q(\tau^{*, 1:I} | \tau^0)} \Big( \sum_{i=0}^{I} \log \phi_\omega(\tau^{*, i}, i) \Big) \right] - \mathbb{E}_{\hat{\tau} \sim \mathcal{D}^P} \left[ \mathbb{E}_{q(\hat{\tau}^{1:I} | \tau^0)} \Big( \sum_{i=0}^{I} \log \phi_\omega(\hat{\tau}^i, i) \Big) \right] \quad (11)$$

where 1) $\mathcal{D}^P$ denotes the nominal trajectory data generated by policy $\pi$ under CMDP $\mathcal{M}_c$ (with the estimated cost before update) and 2) $\phi_\omega(\tau^i, i) = \prod_{t=0}^{T} \phi_\omega(s_t^i, a_t^i, i)$ denotes the permissibility function for accurately determines whether the noise-augmented trajectory $\tau^i$ is feasible. In this manner, the corresponding cost estimation $c_\omega(s_t^i, a_t^i) = -\log \phi_\omega(s_t^i, a_t^i)$ is noise-robust, which can accurately guide the trajectory generation during the diffusion process.

### 3.3 Strategic Exploration for Constraint Update

Based on the above noise-robust objective, we enhance the algorithm's capability for actively discovering feasible constraints by implementing a dynamic exploration algorithm. To achieve this goal, we extend the object (11) by proposing an exploratory constraint update objective designed for strategic exploration, detailed as follows:

$$\mathbb{E}_{\mathcal{D}^E}\left[\mathbb{E}_{q(\tau^{*,1:I}|\tau^0)}\Big(\sum_{i=0}^I \log\phi_\omega(\tau^{*,i},i)\Big)\right] - \mathbb{E}_{\mathcal{D}^P}\left[\mathbb{E}_{q(\hat\tau^{1:I}|\tau^0)}\Big(\sum_{i=0}^I \log\phi_\omega(\hat\tau^i,i)\Big)\right] - \psi(\phi_\omega,\Phi), \quad (12)$$

where $\psi$ refers to a regularization term, controlling the sparsity and the diversity of constraint functions $\phi_\omega$. To learning a constraint representation, motivate by contrastive learning (He et al., 2020), we follow the InfoNCE loss (van den Oord et al., 2018) and implement $\psi(\phi_\omega,\mathcal{Z})$ as:

$$\psi(\phi_\omega,\mathcal{Z}) = \mathbb{E}_{\mathcal{D}^P}\left[\delta\log\frac{e^{\sum_{(s_t,a_t)}\text{dist}[1,\phi_\omega(s_t,a_t)]}}{\sum_{\tilde\phi_\omega\in\mathcal{Z}}e^{\sum_{(s_t,a_t)}\text{dist}[\tilde\phi_\omega(s_t,a_t),\phi_\omega(s_t,a_t)]}}\right], \quad (13)$$

where $\delta$ denotes the regularization parameter controls the scale of sparsity, $\text{dist}(\cdot,\cdot)$ indicates the distance metric, here we choose it to be the $l_1$ norm. $\mathcal{Z}$ denotes the set of feasible constraints that have already been discovered. Intuitively, this regularization term $e^{\sum_{(s_t,a_t)}\text{dist}[1,\phi_\omega(s_t,a_t)]}$ encourages the constraint function to assign higher feasibility values to state-action pairs, improving the sparsity of the inferred constraints. Simultaneously, this objective fosters the diversity of these constraints, allowing them to be distinguishable from previously learned constraints. Since a constraint model can be characterized by its predicted cost values at different state-action pairs, we require different constraint models to assign different feasibility values $\phi_\omega$ to the same state-action pairs

**Implementation.** Based on the above design, Algorithm 1 illustrates the our implementation.

---

**Algorithm 1** Exploratory Inverse Constraint Learning (ExICL)

---

**Require:** Offline dataset $\mathcal{D}^O = \{\mathcal{D}^E,\mathcal{D}^{-E}\}$, contrastive exploration rounds $M$
    Randomly initialize feasibility function $\phi_\omega$ and $\phi^\delta$
    Initialize the set of feasible constraint $\mathcal{Z}\leftarrow\{\emptyset\}$ and the Lagrange parameter $\lambda=0$
    Train diffusion-based trajectory predictor $p_\theta(\tau)$ based on $\mathcal{D}^O$ with objective (5)
    **for** each exploration coefficient $\delta$ **do**
        Initialize the subset of feasible feasibility constraint $\mathcal{Z}^\delta\leftarrow\{\emptyset\}$
        **for** exploration round $m=0,\dots,M$ **do**
            Set the initial feasibility function $\phi_\omega^m=\phi^\delta$
            Initialize the set of predicted trajectories $\mathcal{D}_P=\{\emptyset\}$
            **while** The predicted trajectory $\hat\tau\neq\tau^E,\forall\tau^E\in\mathcal{D}^E$ **do**
                Generate a trajectory $\hat\tau$ with $c(\cdot)=-\log\phi_\omega^m(\cdot)$ as costs and (10) as denoising process.
                Collect the predicted trajectory $\mathcal{D}_P=\mathcal{D}_P\cup\{\hat\tau\}$
                Update $\lambda$ by minimizing the loss $\mathcal{L}=\lambda\mathbb{E}_{\hat\tau\sim\tilde p_{\mathcal{M}_c}}[c(\tau)-\epsilon]$
                Update the feasibility function $\phi_\omega^m$ via objective (12) (based on $\mathcal{D}_P$, $\mathcal{Z}^\delta$ and $\delta$)
            **end while**
            Expand the subset of feasible constraints $\mathcal{Z}^\delta=\mathcal{Z}^\delta\cup\{\phi_\omega^m\}$
        **end for**
        Expand the feasible constraint set $\mathcal{Z}=\mathcal{Z}\cup\mathcal{Z}^\delta$ and reset the feasibility function $\phi^\delta=\phi_\omega^M$
    **end for**

---

## 4 Empirical Evaluation

We empirically evaluate the effectiveness of ExICL by its capability of 1) accurately inferring various types of constraints (e.g., spatial, dynamic, and kinematic) under PointMaze and robot control environments (Section 4.1), 2) exploring various kinds of constraints with strategic exploration in constraint updates (Section 4.2), and 3) accelerating the process of constraint inference (Section 4.3). We empirically evaluate the effectiveness of ExICL by its capability of 1) accurately inferring various types of constraints (e.g., spatial, dynamic, and kinematic) under PointMaze and robot control

environments (Section 4.1), 2) exploring various kinds of constraints with strategic exploration in constraint updates (Section 4.2), and 3) accelerating the process of constraint inference (Section 4.3).

**Experiment Setting.** Our experiments investigate the proposed ExICL in continuous environments. Specifically, we construct three distinct PointMaze environments with different constraints in Figure 2. See Appendix A.1 for further details. To evaluate model performance in more challenging tasks, we extend three robot control environments in MuJoCo (Todorov et al., 2012) by incorporating distinct predefined constraints into different tasks. The examine tasks include 1) *Obstacle HalfCheetah*, where we introduce a *spatial constraint* that prevents the robot from moving

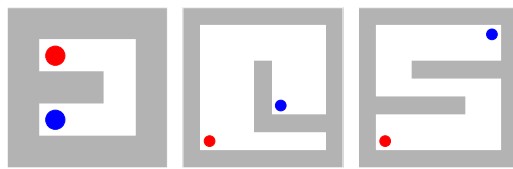

Figure 2: Constrained PointMaze UMAZE, L, and 2WALLS from left to right, where the blue, red, and dark regions indicate the starting, destination, and constrained locations.

backward; 2) *Limited-Speed Walker*, where we design a *dynamic constraint* to control the robot's maximum forward speed; 3) *Blocked Ant*, where we add a *kinematic constraint* on the robot's leg angular velocity, limiting the size of each movement. Please refer to Appendix A.2 for further details. The autonomous experiment setting can be seen in B.3.

**Evaluation Metrics.** To quantify model performance, following (Malik et al., 2021), we use the following evaluation metrics: 1) *Cumulative Reward*, which adds up the total rewards obtained throughout the entire episode; 2) *Cumulative Cost*, which adds the total costs obtained throughout the entire episode and 3) *Feasible Cumulative Reward*, which quantifies the accumulated rewards before any constraint violations occur. Each experiment is repeated with five random seeds, and the results are reported as the mean ± standard deviation (std). The detailed settings and random seeds are reported in Appendix A.

## 4.1 Control Performance: Quantifying the Accurateness of Constraints

In this experiment, we investigate whether ExICL can effectively learn the accurate constraints by assessing whether the learned constraints can facilitate the reproduction of expert demonstrations.

**Constrained PointMaze Environments.** In this experiment, we design three different PointMaze environments, each featuring unique constraints, as shown in Figure 2. The agent's objective is to navigate from the starting location to the target location while successfully avoiding the imposed constraints. As for performance demonstration, we selected the constraints discovered under the largest $\delta$. Since $\delta$ controls the level of regularization on sparsity, our setting is to align with the previous setting of ICRL solvers that favor the sparsity of constraints, thereby providing a fair comparison with previous works.

*Comparison Methods.* We mainly compare the proposed ExICL with the following baselines: 1) Behavior Cloning (**BC**), which learns a policy by directly imitating actions from expert demonstrations; 2) *Least Square Inverse Q-Learning* (**LS-IQ**) (Al-Hafez et al., 2023), which infers reward value functions from offline data to replicate the expert policy. 3) *Inverse Constrained Superior Distribution Correction Estimation* (**ICSDICE**) (Quan et al., 2024), which utilizes reward information and solves a regularized dual optimization problem for safe control by exploiting the dataset.

Table 1: PointMaze evaluation performance. Each value is reported as the mean ± std over 100 runs and 5 seeds. We highlight the best results with the highest rewards or lowest violations in bold.

| Methods | PointMaze-UMAZE | | | PointMaze-2WALLS | | | PointMaze-L | | |
|---|---|---|---|---|---|---|---|---|---|
| | Reward ↑ | Cost ↓ | Reward w/o Cost ↑ | Reward ↑ | Cost ↓ | Reward w/o Cost ↑ | Reward ↑ | Cost ↓ | Reward w/o Cost ↑ |
| BC | $0.88 \pm 0.10$ | $2.09 \pm 5.03$ | $0.78 \pm 0.10$ | $0.90 \pm 0.16$ | $3.29 \pm 5.04$ | $0.66 \pm 0.14$ | $0.92 \pm 0.05$ | $1.15 \pm 3.71$ | $0.92 \pm 0.05$ |
| LS-IQ | $0.84 \pm 0.16$ | $10.28 \pm 7.68$ | $0.54 \pm 0.20$ | $0.82 \pm 0.24$ | $22.36 \pm 9.28$ | $0.58 \pm 0.28$ | $0.76 \pm 00.10$ | $14.97 \pm 5.63$ | $0.48 \pm 0.12$ |
| ICSDICE | $\mathbf{1.00 \pm 0.00}$ | $0.02 \pm 0.01$ | $0.95 \pm 0.01$ | $0.94 \pm 0.06$ | $1.20 \pm 0.06$ | $0.78 \pm 0.10$ | $0.92 \pm 0.04$ | $0.14 \pm 0.05$ | $0.62 \pm 0.08$ |
| ExICL (ours) | $\mathbf{1.00 \pm 0.00}$ | $\mathbf{0.01 \pm 0.00}$ | $\mathbf{0.99 \pm 0.00}$ | $\mathbf{1.0 \pm 0.00}$ | $\mathbf{0.01 \pm 0.00}$ | $\mathbf{0.99 \pm 0.01}$ | $\mathbf{1.0 \pm 0.00}$ | $\mathbf{0.00 \pm 0.00}$ | $\mathbf{1.0 \pm 0.00}$ |

*Results Analysis.* Table 1 presents our evaluation results, revealing that ExICL consistently outperforms other baselines by achieving higher cumulative rewards and lower rates of constraint violations. A key factor contributing to this performance is that classic methods such as LS-IQ and BC do not intentionally model the constraint. Consequently, there is no guarantee that the agent

will maintain a safe distance from the wall, often resulting in substantial costs and reduced feasible rewards.

*Visualization.* To better illustrate the learned constraint, Figure 3 shows the validation of trajectory level constraints. We find that ExICL accurately captures the feasibility of trajectories.

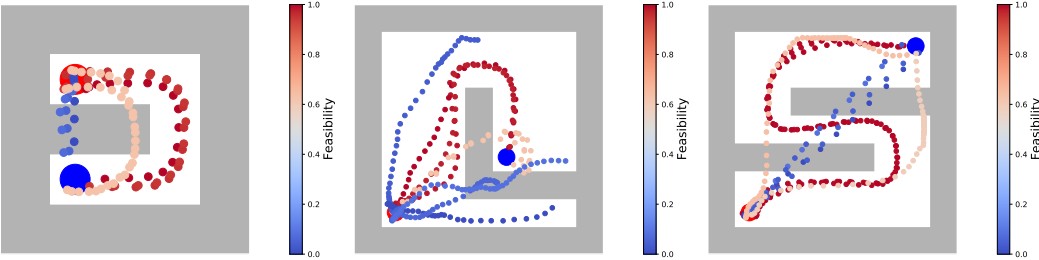

Figure 3: Visualization of the constraints learned by ExICL under the PointMaze-UMAZE, 2WALLS, and L environments. Each point indicates the predicted cost within a trajectory

**Constrained MuJoCo Environments.** Three different MuJuCo environments that differ in agent and constraint are designed for this experiment. The agent's objective is to cover the maximum possible distance within a unit of time while prevent violating the constraints.

*Comparison Methods.* To perform a more comprehensive evaluation, we add more baselines to previously compared ones: 1) **SMODICE** (Ma et al., 2022), which leverages the dual and offline reward function to optimize the policy; 2) **OptiDICE-Constraint**, which replaces the DICE objective used in ICSDICE with OptiDICE (Lee et al., 2021); and 3) **SMODICE-Constraint**, which incorporates information by adding environment rewards to the SMODICE's learned discriminator reward.

Table 2: MuJoCo evaluation results. The baseline results are adapted from (Quan et al., 2024). Each value is reported as the mean $\pm$ std over 10 runs and 5 seeds. **Bold** denotes safe methods with maximum rewards.

| Methods | Obstacle-HalfCheetah | | Limited-Walker | | Blocked-Ant | |
|---|---|---|---|---|---|---|
| | Reward w/o Cost $\uparrow$ | Cost $\downarrow$ | Reward w/o Cost $\uparrow$ | Cost $\downarrow$ | Reward w/o Cost $\uparrow$ | Cost $\downarrow$ |
| BC | $731 \pm 693$ | $0.30 \pm 0.40$ | $-7 \pm 0.5$ | $0.01 \pm 0.02$ | $876 \pm 138$ | $0.04 \pm 0.02$ |
| LS-IQ | $2175 \pm 775$ | $0.04 \pm 0.05$ | $603 \pm 203$ | $0.17 \pm 0.09$ | $-63 \pm 208$ | $0.40 \pm 0.12$ |
| SMODICE | $3565 \pm 345$ | $0.13 \pm 0.11$ | $2334 \pm 238$ | $0.52 \pm 0.19$ | $1410 \pm 153$ | $0.33 \pm 0.07$ |
| SMODICE-c | $3829 \pm 661$ | $0.30 \pm 0.24$ | $1871 \pm 155$ | $0.18 \pm 0.15$ | $1763 \pm 180$ | $0.53 \pm 0.03$ |
| OptiDICE-c | $2749 \pm 597$ | $0.03 \pm 0.06$ | $1538 \pm 283$ | $0.01 \pm 0.01$ | $3070 \pm 91$ | $0.01 \pm 0.00$ |
| ICSDICE | $2315 \pm 740$ | $0.04 \pm 0.04$ | $1587 \pm 308$ | $0.01 \pm 0.02$ | $\mathbf{3073 \pm 103}$ | $\mathbf{0.01 \pm 0.00}$ |
| ExICL (ours) | $\mathbf{5298 \pm 480}$ | $\mathbf{0.00 \pm 0.00}$ | $\mathbf{1862 \pm 29}$ | $\mathbf{0.00 \pm 0.00}$ | $3061 \pm 199$ | $0.01 \pm 0.01$ |

*Results Analysis.* Table 2 shows the evaluation results in high-dimensional robot control tasks. We can find that ExICL consistently achieves fewer constraint violations across all three environments, regardless of the constraint type, demonstrating the effectiveness of the inferred constants. In contrast, offline IL methods, such as BC, IS-IQ, and SMODICE, generally fail to ensure safety, even when reward information is incorporated. This limitation is expected, as these methods are not explicitly designed to address safety concerns. Interestingly, OptiDICE-c, with its normalization and soft-chi divergence tricks, and ICSDICE, utilizing a superior DICE approach tailored for constraint learning, exhibit satisfactory performance with low costs and relatively high rewards. However, neither method surpasses ExICL in terms of rewards, except in the Ant environment where performance is comparable. This is especially evident in the HalfCheetah environment, where ExICL's exploration ability enables the agent to more effectively pursue rewards.

## 4.2 EXPLORATORY PERFORMANCE: EVALUATING THE DIVERSITY OF CONSTRAINTS

In this section, we study the exploratory performance of our method by quantifying the diversity of constraints within the learned feasible constraint set.

**Constrained PointMaze Environments.** In this experiment, we study whether the ExICL can learn a diverse set of constraints by analyzing the cost values assigned to the expert trajectories and non-

expert ones. Ideally, the constraints in the feasible set should assign different costs to the same trajectory, in the meantime, the expert trajectories should consistently have lower costs.

*Comparison Methods.* In this study, we primarily compare several common exploration methods, including 1) Random-Noise, which involves directly adding noise to the parameters of learned constraints to discover alternative constraints; and 2) Random Initialization, where we repeatedly learn different constraints starting from normal distributed initial model parameters other than zero constant initialization for other methods.

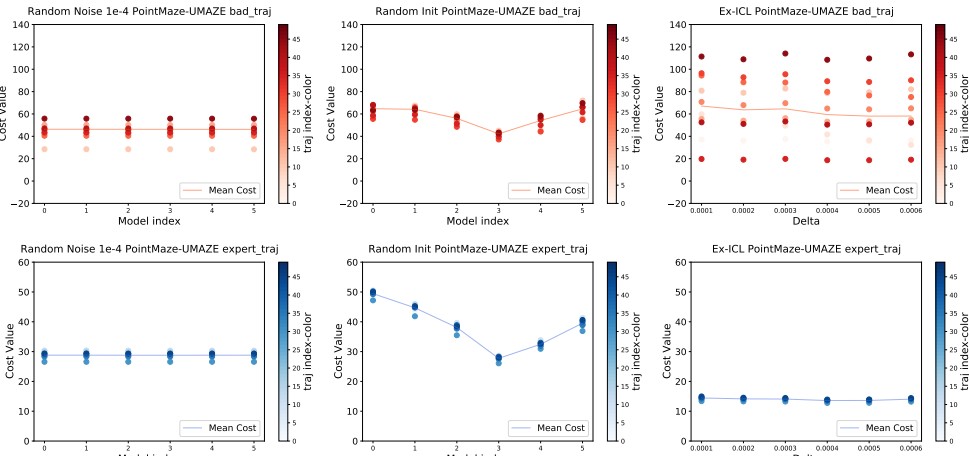

Figure 4: The delta varying exploratory results for both constraint-violating trajectories (top) and expert trajectories (bottom) of PointMaze-UMAZE environment. Each data point corresponds to the cumulative costs for a trajectory. Three exploration strategies are presented: random noise (left), random initialization (middle) and Ex-ICL (right).

Figure 4 illustrates the exploratory results in the PointMaze-UMAZE environment, with exploratory models varying in regularization parameter $\delta$. Additional $\delta$-varying exploratory results for the other two environments, PointMaze-L and PointMaze-2Walls, are displayed in Figure 7 and Figure 8 in the Appendix, and the exploratory results depending on exploration rounds are displayed in Figure 9. The decrease of cost value as regularization parameter $\delta$ increase in Figure 4, 7, 8 implies the enlarging sparsity of the constraints. Our results reveal that the diversity of feasible constraints discovered by random noise and initialization methods is less effective compared to that achieved by our ExICL method. Another intriguing observation is the costs of constraint-violating trajectories have a significantly higher variance than those of expert trajectories. This is because, to recover the constraint, ICRL algorithms must increase the cost values of bad trajectories above the threshold $\epsilon$. On the other hand, for the cost values of expert trajectories, ICRL algorithms must guarantee their values to be smaller than $\epsilon$. In this work, our $\epsilon$ is set to close to zero, so the scale of variances for bad trajectories is much larger than those of expert trajectories. By implementing strategic exploration, our EX-ICL exploration strategy successfully identifies this diverse set of feasible cost models, causing the variance of predicted cost values to be higher.

**Robot Control Tasks.** Figure 5 visualizes the robot trajectory segments during evaluation in three MuJoCo environments. From left to right are the results without constraint model (nominal), and with three different constraint models during exploration. The constrained value of the last frame is displayed, where yellow indicates constraint violations and green indicates safety. We present the average reward and cost per step for each figure. We observe that different constraint models lead to distinct behaviors, with red circles highlighting the key differences. For example, in the HalfCheetah environment, the robot moves varying distances due to the exploration process influenced by the constraint model. Similarly, the Walker robot is encouraged to lift its foot higher to increase speed, while the Ant robot explores in the northward direction. Appendix B.2 reports numerical results.

### 4.3 LEARNING EFFICIENCY: HOW FAST A CONSTRAINT CAN BE INFERRED

We conduct an empirical study to compare the learning efficiency of our ExICL algorithm with other baselines within the PointMaze environment. The learning efficiency is measured by data points used for training as emphasized in offline reinforcement learning where the dataset collection

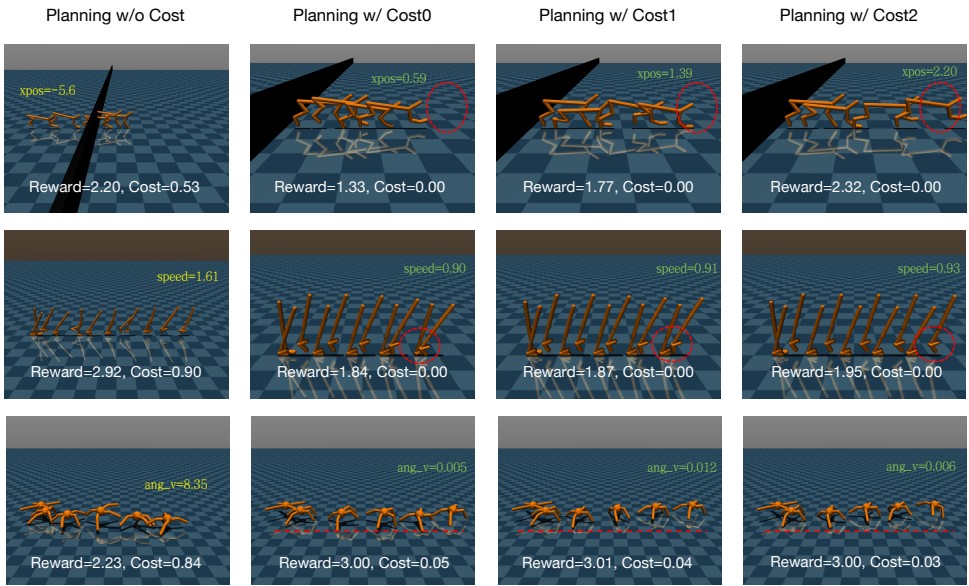

Figure 5: Visualization of the exploration results in MuJoCo environments. Each row represents an environment with identical frames for comparison.

is more costly (Levine et al., 2020). The data efficiency curve illustrated in Figure 6 below depicts the trend of performance as data point usage increases. The EX-ICL curve represents the progression till the discovery of the first valid constraint, aiming to offer a fair comparison with previous work that studies only one constraint. The area circled with an orange ellipse represents the training phase of the GDV model and reward value model. Among the compared methods, we find that our ExICL exhibits the highest sample efficiency. This efficiency stems from its ability to learn feasible constraints that yield significant rewards while utilizing a minimal number of data points. This is attributed to our GDV model, which performs a generative denoising process to validate candidate constraints without relying on frequent policy updates, thereby significantly improving sample complexity.

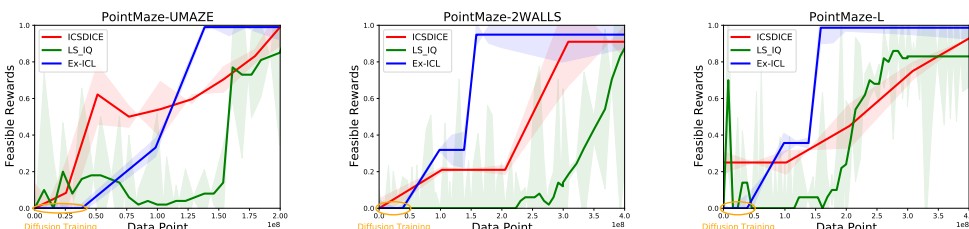

Figure 6: Training efficiency of three methods in three PointMaze environments.

## 5 CONCLUSION

In the paper, we introduced the ExICL algorithm, which is designed to learn a diverse set of constraints from an offline demonstration dataset. By proposing a GDV model, we significantly accelerated the verification of candidate constraints. Additionally, we developed a strategic exploration mechanism that updates constraints and efficiently expands the constraint set, thereby facilitating the discovery of varied constraints. To validate our method, we conducted experiments across a diverse array of tasks, including navigation, locomotion, and autonomous driving, and under various types of constraints. Our results demonstrate that ExICL significantly outperforms other baseline methods in terms of learning more accurate constraints, discovering diverse constraints, and enhancing learning efficiency. A promising direction for future work involves expanding ExICL to more practical environments, such as quadrapedal and humanoid robot control.

ACKNOWLEDGMENTS

This work is supported in part by Shenzhen Science and Technology Major Program under grant KJZD20240903104008012, Shenzhen Fundamental Research Program (General Program) under grant JCYJ20230807114202005, Guangdong-Shenzhen Joint Research Fund under grant 2023A1515110617, Guangdong Basic and Applied Basic Research Foundation under grant 2024A1515012103, and Guangdong Provincial Key Laboratory of Mathematical Foundations for Artificial Intelligence (2023B1212010001).

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

## A    IMPLEMENTATION AND ENVIRONMENTAL DETAILS

### A.1    POINTMAZE ENVIRONMENTS

We designed three PointMaze environments: one with a grid of $5 \times 5$ cells and two others with $7 \times 7$ grids. Each cell spans an area of $0.5\,\mathrm{m} \times 0.5\,\mathrm{m}$. The center of the grid is positioned at the origin $(0, 0)$. Constraints are applied at specific cells marked as "1" on the corresponding maze map. The outer walls, also marked as "1", are impassable, while the inner regions allow the agent to move freely.

The agent is modeled as a 2-DoF ball actuated by forces in the Cartesian $x$- and $y$-directions. The agent's objective is to navigate through the maze and reach a designated target. The goal is considered reached when the Euclidean distance between the ball and the target is less than $0.45\,\mathrm{m}$. The reward system assigns a value of 1 when the agent reaches the goal, while all other cells have a reward of 0. Similarly, entering a constrained location incurs a cost of 1. The environment terminates after a maximum of 150, 500, or 250 time steps, depending on the specific maze.

The state space is continuous and comprises four dimensions: the agent's $x$ and $y$ coordinates, as well as the linear velocity in both directions. The action space is also continuous, consisting of accelerations in the $x$- and $y$-axes.

### A.2    MUJOCO ENVIRONMENTS

Our simulated environments are constructed using Mujoco. We derived state-action function $c(s, a)$ from a state-dependent function $c(s')$ with $c(s, a) = \mathbb{E}_{s' \sim P(s'|s,a)}[c(s')]$. Below are more details regarding the environments used:

1. **HalfCheetah (Obstacle)**: These environments are adapted from Liu et al. (2023). In this setup, the agent controls a robot that moves faster backward than forward. The rewards are based on the distance covered between consecutive time steps, along with penalties tied to the magnitude of the actions. Additionally, a constraint restricts movement to areas where the X-coordinate is greater than -3, forcing the robot to move forward only.

2. **Ant (Blocked)**: In these environments, the agent manages a robot that moves forward and gains rewards based on the distance traveled. However, a constraint limits the robot's leg angular velocity to prevent excessive force on the ground. The limit is set at 1.

3. **Walker (LS)**: In these environments, the agent controls a robot that moves forward and earns rewards for traveling distances. However, there is a speed limit is set at 1, resulting in reduced rewards compared to an environment without such a constraint.

### A.3    OFFLINE DATASETS

**Offline Dataset for Navigation Tasks.** This offline dataset collects a total number of 3255 trajectories consisting of around $1.5 \times 10^5$ state-action pairs in three environments. More precisely, we collect 1638 trajectories for PointMaze-UMAZE, 667 trajectories for PointMaze-2Walls, and 950 trajectories for PointMaze-L. In each individual environment, the trajectories can be categorized into three parts: 1) expert trajectories generated by the expert policy trained under the PPO-Lagrangian algorithm and incorporates stochasticity of $0.05$, allowing for random actions; 2) constraint-violating trajectories created by a policy that accelerates the agent's movement directly toward the terminating location, with stochasticity of $0.1$; 3) random trajectories generated by the uniformly random policy. The proportion of the number of pairs in each kind of trajectory is around $5:1:1$.

**Offline Dataset for Robot Control Tasks.** We use the public offline dataset provided by (Quan et al., 2024). Specifically, this offline dataset includes a total number of 250 trajectories, which obtains 200 suboptimal trajectories (each with 1000 steps) and 50 expert trajectories from a PPO-lag algorithm.

### A.4    MODEL ARCHITECTURES

We construct our generative diffusion verifier and reward value model following the official implementation of (Janner et al., 2022), both with a U-Net based architecture. We also design our

cost value model similar to the reward value model with a U-Net based architecture but differs in outputting a horizon $H$-length feature takes value in $[0,1]$ representing the feasibility $\phi_\omega(s_t, a_t)$ for each state-action pair in the $H$-length trajectory. And the cost value is explicitly calculated by $V_c = \sum_{t=0}^{H} \gamma^t c_\omega(s_t^i, a_t^i, i) = \sum_{t=0}^{H} \gamma^t - \log \phi_\omega(s_t^i, a_t^i, i)$. Note that diffusion time $i$ is explicit input into the network and embedded by an MLP to perceive the denoising process. Thus the cost value model can accurately predict the cost value of noisy trajectory and use it to guide the generation. Environment-dependent model hyperparameters are presented in 3 below.

## A.5 EVALUATION DETAILS

The evaluation in MuJoCo is conducted over 10 runs using 5 random seeds, while the evaluation in the PointMaze environments is carried out over 100 runs with 5 random seeds.

Table 3: List of the utilized hyperparameters in the navigation tasks in PointMaze and MuJoCo environments.

| Parameters | PointMaze-UMAZE | PointMaze-2WALLS | PointMaze-L | Obstacle-HalfCheetah | Limitted-Walker | Blocked-Ant | CommonRoad-Velocity<40 |
|---|---|---|---|---|---|---|---|
| Max Episode Length | 150 | 500 | 250 | 1000 | 1000 | 1000 | 400 |
| Discount Factor | 0.99 | 0.99 | 0.99 | 0.99 | 0.99 | 0.99 | 0.99 |
| Episodes Collected | 64 | 64 | 64 | 64 | 64 | 64 | 64 |
| Policy Batchsize | 512 | 1024 | 1024 | 1024 | 1024 | 1024 | 1024 |
| Expert Batchsize | 512 | 1024 | 1024 | 1024 | 1024 | 1024 | 1024 |
| Initial Lagrange Multiplier | 10 | 100 | 100 | 5 | 5 | 5 | 200 |
| Lagrange Multiplier Learning Rate | 0.01 | 0.01 | 0.01 | 0.1 | 0.1 | 0.1 | 0.1 |
| Guided Scale | 0.1 | 0.01 | 0.001 | 0.1 | 0.01 | 0.1 | 0.01 |
| Cost Model Horizon | 32 | 32 | 32 | 32 | 32 | 32 | 32 |
| Cost Model Learning Rate | 1e-4 | 1e-5 | 1e-5 | 1e-5 | 1e-5 | 1e-5 | 5e-5 |
| Cost Model Update Step | 4 | 1 | 4 | 4 | 4 | 4 | 2 |
| Diffusion steps | 20 | 20 | 20 | 20 | 20 | 20 | 20 |
| Diffusion Time Feature Dimension | 32 | 32 | 32 | 32 | 32 | 32 | 32 |
| Diffusion Time Hidden Dimension | 128 | 128 | 128 | 128 | 128 | 128 | 128 |
| Hidden Feature Dimension | 32 | 32 | 32 | 32 | 32 | 32 | 32 |
| Convolution Kernal Size | 5 | 5 | 5 | 5 | 5 | 5 | 5 |
| U-Net depth | 4 | 4 | 4 | 4 | 4 | 4 | 3 |
| Convolution Layers Dimension | (1×,2×,4×,8×) | (1×,2×,4×,8×) | (1×,2×,4×,8×) | (1×,2×,4×,8×) | (1×,2×,4×,8×) | (1×,2×,4×,8×) | (1×,4×,8×) |

# B ADDTIONAL RESULTS

## B.1 ADDITIONAL EXPLORATORY RESULTS OF THE CONSTRAINTS IN POINTMAZE ENVIRONMENT

The following 2 figures illustrate the output cost value of the cost models trained under different exploratory regularization parameters $\delta$ for PointMaze-L and PointMaze-2-Walls environments respectively.

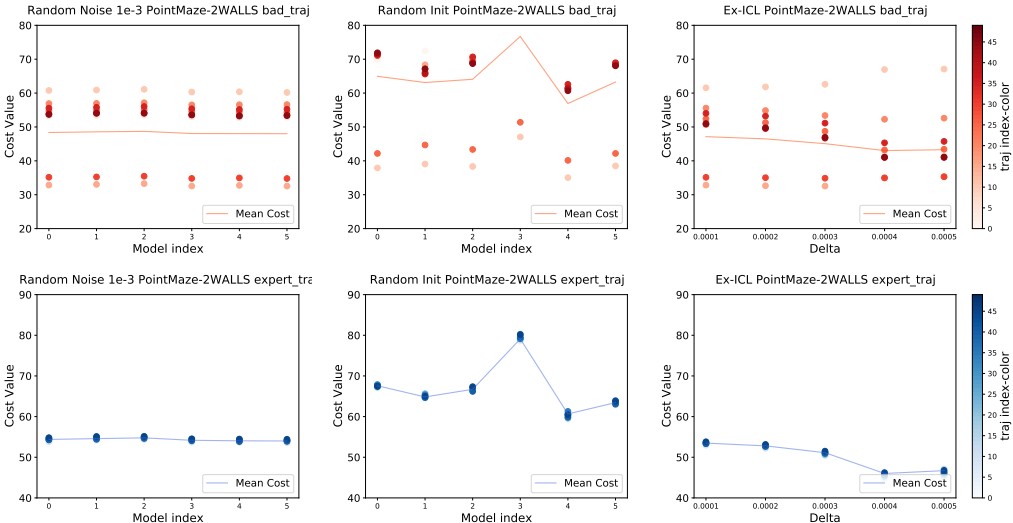

Figure 7: The delta varying exploratory results for both bad trajectories (top) and expert trajectories (bottom) of PointMaze-2WALLS environment. Each data point corresponds to the cumulative costs for a trajectory. Three exploration strategies are presented: random noise (left), random initialization (middle) and Ex-ICL (right).

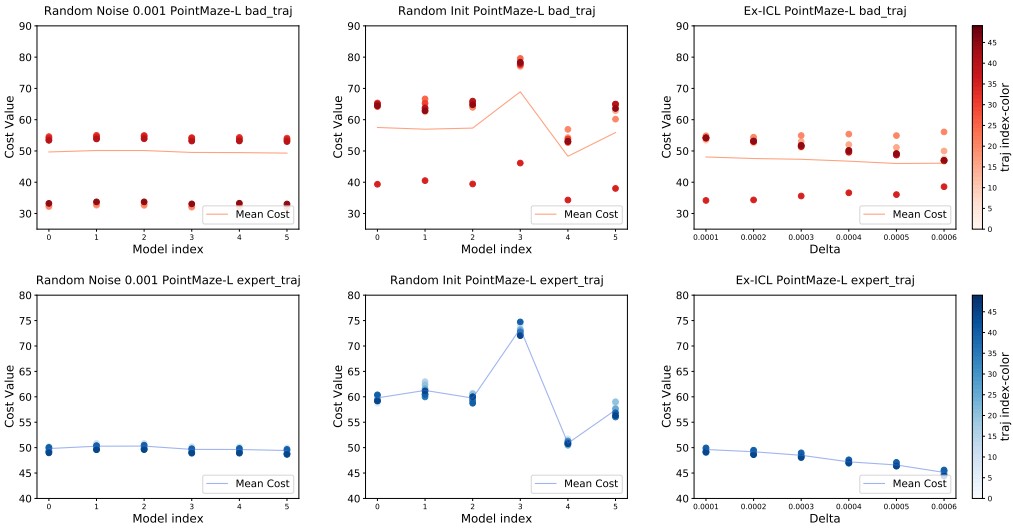

Figure 8: The delta varying exploratory results for both bad trajectories (top) and expert trajectories (bottom) of PointMaze-L environment. Each data point corresponds to the cumulative costs for a trajectory. Three exploration strategies are presented: random noise (left), random initialization (middle), and Ex-ICL (right)

We also illustrate the cost value-exploration round figures below, in which the cost model is trained under the largest $\delta$ to demonstrate the effectiveness of our exploration method:

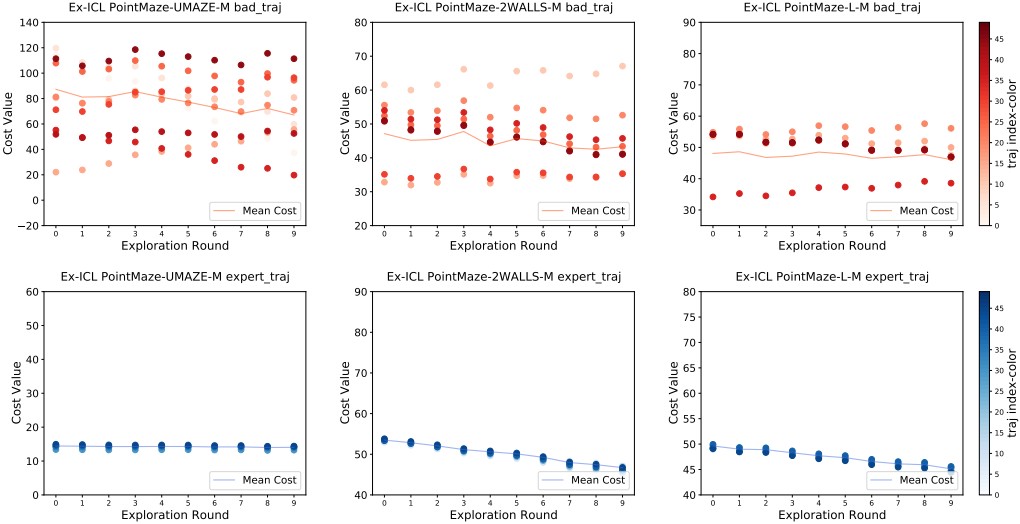

Figure 9: The exploration round varying exploratory results for both bad trajectories (top) and expert trajectories (bottom) of PointMaze-UMAZE, PointMaze-2WALLS, and PointMaze-L environments. Each data point corresponds to the cumulative costs for a trajectory. Three exploration strategies are presented: random noise (left), random initialization (middle) and Ex-ICL (right).

As we expected, the cost values of the same trajectory gradually vary from the original ones, as the InfoNCE term in 13 encourages them to be distinct.

## B.2 NUMERICAL ANALYSIS FOR THE DIVERSITY OF THE CONSTRAINT IN MUJOCO ENVIRONMENT

For numerical analysis, we use five different constraint models during exploration to estimate the cost values of identical trajectory segments in each environment and report the mean and standard deviation results across them. The results, shown in Table 4, indicate that: 1) constraint models effectively distinguish between safe and violated segments, assigning higher

Table 4: Costs values across varying exploration epochs. The reported values are the mean $\pm$ std over 5 different constraints.

| Environment | HalfCheetah | Walker | Ant |
|---|---|---|---|
| **Violated Segments** | 49.2±5.7 | 23.9±2.2 | 36.6±5.4 |
| **Safe Segments** | 6.1±0.7 | 10.9±0.8 | 10.1±2.1 |

cost values to violated trajectories and lower values to safe ones; and 2) the std results across the five constraint models suggest that the exploration process successfully induces variability among the constraint models.

## B.3 ADDITIONAL PERFORMANCE EXPERIMENTS IN COMMONROAD ENVIRONMENT WITH HIGHD DATASET

To demonstrate the effectiveness of our method in a more realistic environment, we conducted experiments on performance in CommonRoad-RL (Wang et al., 2021) Environment with a velocity<40 constraints. We chose the processed HighD (Krajewski et al., 2018) data given by (Liu et al., 2023), aligning with baseline method (Quan et al., 2024). Our methods outperformed the baseline methods in terms of trajectory cumulative cost>0 rate while achieving comparable performance on cumulative reward and AUC metric. Such a result suggested that our method maintains effectiveness in complicated realistic environments and broadens its future applications.

Table 5: Performance measured by reward, cost, and Area Under Curve (AUC) of different methods. The reported value of reward and cost are the mean $\pm$ std over 500 runs under 5 random seeds

| Methods | BC | OptiDice-c | ICSDICE | EX-ICL(ours) | Expert |
|---------|-----|-----------|---------|--------------|--------|
| reward | -2.3$\pm$2.3 | -1.6$\pm$3.7 | 10.7$\pm$1.5 | 10.5$\pm$2.2 | 14.0 |
| cost | 1.4% | 12% | 6.5% | **3.2%** | 0.8% |
| AUC | NA | 0.76$\pm$0.02 | 0.81$\pm$0.01 | 0.84$\pm$0.06 | NA |

## C  PERFORMANCE OF DEMONSTRATIONS, BASELINE, AND OUR METHOD

To support our claim that our method achieved expert-level performance in all 3 MuJoCo locomotion environments and thus the large or small performance gaps between our method and baseline (Quan et al., 2024) are determined by the gap between expert demonstration and baseline performance, we reported the cumulative feasible reward and the cumulative cost of expert demonstration, suboptimal demonstration, our method, and baseline (Quan et al., 2024) in the below Table 6.

Table 6: MuJoCo demonstration and policy cumulative feasible reward and cumulative cost. **Bold** denotes expert demonstration its comparable performances .

| Methods | Obstacle-HalfCheetah | | Limited-Walker | | Blocked-Ant | |
|---------|----------------------|----------|----------------|----------|-------------|----------|
| | Reward w/o Cost $\uparrow$ | Cost $\downarrow$ | Reward w/o Cost $\uparrow$ | Cost $\downarrow$ | Reward w/o Cost $\uparrow$ | Cost $\downarrow$ |
| Expert | **4915 $\pm$ 1170** | **0.00 $\pm$ 0.00** | **1870 $\pm$ 12** | **1.64 $\pm$ 1.67** | **3059 $\pm$ 276** | **7.22 $\pm$ 2.74** |
| Suboptimal | 3784 $\pm$ 1455 | 487 $\pm$ 482 | 1003 $\pm$ 757 | 309 $\pm$ 243 | 732 $\pm$ 595 | 233 $\pm$ 213 |
| ICSDICE | 2315 $\pm$ 740 | 0.04 $\pm$ 0.04 | **1587 $\pm$ 308** | **0.01 $\pm$ 0.02** | **3073 $\pm$ 103** | **0.01 $\pm$ 0.00** |
| ExICL (ours) | **5298 $\pm$ 480** | **0.00 $\pm$ 0.00** | **1862 $\pm$ 29** | **0.00 $\pm$ 0.00** | **3061 $\pm$ 199** | **0.01 $\pm$ 0.01** |

## D  RELATED WORKS

In this section, we introduce the previous works that are most related to our approach.

**Inverse Constraint Learning.** Inverse Constraint Learning (ICL) aims at recovering constraints encoded in demonstrations to autonomously define and reuse constraints. The ICL problem is inherently ill-posed since there can be multiple combinations of reward and constraint pairs that explain the optimality of expert behaviors. To address this ambiguity, conventional ICL methods assumed the constraints are generated from certain constraint templates(Chou et al., 2021; Park et al., 2020; Pérez-D'Arpino & Shah, 2017), while recent approaches leveraged neural networks to represent constraints over discrete state-action spaces (Scobee & Sastry, 2020) and continuous state-action spaces (Malik et al., 2021; Liu et al., 2023). Continuous works explore ICL under the environment with soft constraint (Garg et al., 2021), multiple agents (Liu & Zhu, 2022; 2024; Qiao et al., 2023), stochastic transition dynamics (McPherson et al., 2021; Xu & Liu, 2023), multiple tasks (Kim et al., 2023) and robust optimization framework (Xu & Liu, 2024). A recent study (Quan et al., 2024) has explored the method of ICL from offline dataset. However, these methods mainly focus on identifying a specific constraint without exploring the approach to infer a diverse set of constraints.

**Diffusion Planner for RL.** Due to the impressive generative capabilities, some recent studies have explored using diffusion models as planners to simulate environmental dynamics and solve RL problems in a model-based manner (Zhu et al., 2023). Among these studies, Diffuser (Janner et al., 2022) guided the denoising process in diffusion models using a probabilistic representation of an RL policy. To further enhance model performance, subsequent research has expanded on this concept. AdaptDiffuser (Liang et al., 2023) incorporated evolutionary planning, while EDGI (Brehmer et al., 2023) focused on planning by leveraging the geometric structure in the task. Additionally, studies such as MTDiff (He et al., 2023) explored multi-task planning, TCD (Hu et al., 2023) utilized temporal information for guiding the planning, LatentDiffuser (Li, 2024) examined planning in latent action spaces, (Ni et al., 2023) studied the diffusion planning under meta RL setting and HDMI (Li et al., 2023) developed a hierarchical decision framework for diffusion policy. In the realm of safety-critical applications, while recent studies such as SafeDiffuser (Xiao et al., 2023) and LTLDoG(Feng et al., 2024) have proposed safe planning using diffusion models under given constraints, the extension of this method to the inverse RL problem remains unexplored.

