# OpenReview forum: "Toward Exploratory Inverse Constraint Inference with Generative Diffusion Verifiers"
_ICLR.cc/2025/Conference — ICLR 2025 Poster_

### Official Review · Reviewer_56R9 · 2024-10-30

**Soundness:** 3
**Presentation:** 3
**Contribution:** 3
**Rating:** 6
**Confidence:** 3

**Summary:**

The authors consider inverse constraint learning, and improve on previous work by constructing an algorithm that can generate a set of constraints, and verify those constraints by applying techniques developed in diffusion modelling for RL. In particular, the authors construct a guidance term that is the gradient of a set of feasiblity terms, which they can use for on the fly verification of the proposed feasilibity functions, thereby eliminating a costly second optimization loop. The authors test their proposed algorithm on a variety of RL benchmarks.

**Strengths:**

- The technique makes clever use of the advanatages found in diffusion techniques: being able to modify the policy at run time by applying guidance terms
- The paper strikes a good balance of building a new method out of existing elements.

**Weaknesses:**

The authors seem to omit some details of their mechanism, which I think are quite crucial to the paper. These are:
- how is reward treated? Is a separate reward model that is (1) differentiable, and (2) conditioned on diffusion time (i in the author's notation) trained following Janner et al? These details are not present in Alg. 1, but are necessary to evaluate the gradient p_Mc in eqns (9) and (10).
- It is also not made clear whether in (9) and (10) the feasibility functions and reward are made to condition on diffusion time i, as I would expect it should since only tau_i is available at i.
- After algorithm 1 is complete, how is the final policy constructed for the experiments? Perhaps this is as simple as running eqn. (9) and (10) a final time, but this is not specified either.
- After algorithm 1 completes, how are constraints chosen by the practitioner as the abstract says? How do the authors choose what constraints they apply when sampling their final evaluations? This is stated in the abstract but is not discussed in the paper at all.
- How is constrained data collected? Is there an expert that already includes the constraint?

Minor:
- A few scattered grammar errors could be addressed

**Questions:**

See also questions under "weaknesses"
- Do the authors have some intuition why their method seems to outperform baselines significantly for HalfCheetah, marginally for Limited-Walker and only ties for Blocked-Ant?
- In the MuJoCo experiments, is the reward presented in Table 2 the feasible reward? I.e. are rewards truncated after a constraint has been violated? It seems that that would be the more inveresting metric to report, I would recommend the authors report that metric, and if they already do so make it clear it is that metric.

---

> ### Author Response · Authors · 2024-11-22
> **Author response to Reviewer 56R9 - Part 1**
>
> &nbsp;*1. how is reward treated? Is a separate reward model that is (1) differentiable, and (2) conditioned on diffusion time (i in the author's notation) trained following Janner et al? These details are not present in Alg. 1, but are necessary to evaluate the gradient $p_\mathcal{M}^c$ in eqns (9) and (10).*
>
> **Response.** Thank you for pointing out this important detail. Yes, our reward model follows the design of [1] Janner et al., employing a U-Net based neural network architecture. This network takes as input a horizon $H$-length noisy trajectory and the diffusion timestep, $i$, to predict the corresponding reward value. Its training is supervised, using noisy trajectories and their associated rewards. As a differentiable neural network, it provides the necessary gradients for classifier-free guidance of guiding the generative diffusion verifier. This design choice and its details are now clearly explained in Section A.4 of the Appendix in the revised manuscript.
>
> &nbsp;*2. 'It is also not made clear whether in (9) and (10) the feasibility functions and reward are made to condition on diffusion time i, as I would expect it should since only $\tau_i$ is available at i.'*
>
> **Response.** Thanks for pointing out this concern. You are correct; both the feasibility functions and the reward model are conditioned on the diffusion timestep, $i$. This conditioning is essential because these models must be aware of the current stage of the denoising process to effectively guide the generative diffusion model. In our implementation, both the reward model and the feasibility model process the input diffusion timestep, $i$, using a Multilayer Perceptron (MLP) layer. The output of this MLP is then broadcast-added to the input trajectory, following the approach described in [1] Janner et al. This detail has been added to Section A.4 of the Appendix in the revised manuscript.
>
> &nbsp;*3. 'After algorithm 1 completes, how are constraints chosen by the practitioner as the abstract says?'*
>
> **Response.** Thanks for raising this important point. As shown in Algorithm 1, each discovered constraint is conditioned on a specific regularization parameter $\delta$. This parameter directly controls the level of sparsity of the discovered constraint, with larger values of $\delta$ encouraging the feasibility $\phi$ to be closer to 1. The choice of constraint is thus directly influenced by the practitioner's preference for sparsity. Specifically, a sparse constraint minimizes its impact on the reward-maximizing policy, as demonstrated in Section 5.1 of [3]. Conversely, a dense constraint offers stronger safety guarantees but at the cost of potentially more significant restrictions on the agent's actions and a reduction in the overall reward.
>
> &nbsp;*4. 'How do the authors choose what constraints they apply when sampling their final evaluations? This is stated in the abstract but is not discussed in the paper at all.'*
>
> **Response.** We appreciate you pointing out this important concern.
>
> Our experiments evaluate "Control Performance" in Section 5.1, "Exploratory Performance" in Section 5.2, and "Learning Efficiency" in Section 5.3. The respective choice of constraint in each experiment is that:
>
> 1) For control performance (section 5.1), we selected the constraints discovered under the largest $\delta$. Since $\delta$ controls the level of regularization on sparsity, our setting is to align with the previous setting of ICRL solvers that favor the sparsity of constraints, such as Section 5.1 of [1], thereby providing a fair comparison with previous works.
>
> 2) For exploratory performance (Section 5.2), all the constraints stored in the constraint pool are included in the evaluation, with which we illustrated the diversity of learned constraints in Figure 4, 7, 8, and 9.
>
> 3) In terms of learning efficiency (Section 5.3), we illustrate the sample complexity in discovering the first valid constraint with EX-ICL. This is to offer a fair comparison with previous work that study only one constraint.
>
> We have clarified these in the revised version of our paper.
>
> &nbsp;*5. 'How is constrained data collected? Is there an expert that already includes the constraint?'*
>
> **Response.** We appreciate you raising this consideration. The constraint-satisfying data is generated by an expert policy trained under a ground-truth constraint. This approach is consistent with many previous ICRL studies [2]. However, due to the ill-posed nature of the Inverse Constrained Reinforcement Learning (ICRL) problem, the ground-truth constraint is not uniquely identifiable from expert demonstrations alone. Therefore, our method employs exploration to identify a feasible set of constraints, providing flexibility and efficiency to constraint inference.

---

> ### Author Response · Authors · 2024-11-22
> **Author response to Reviewer 56R9 - Part 2**
>
> &nbsp;*6. 'Do the authors have some intuition why their method seems to outperform baselines significantly for HalfCheetah, marginally for Limited-Walker and only ties for Blocked-Ant?'*
>
> **Response.** Thank you for raising this important concern. In fact, our methods have achieved comparable performance to the expert agent across all 3 environments. The significant lead of HalfCheetah is because other baselines having difficulty in resolving the HalfCheetah environment.
>
> We have reported the reward and cost performance of our method, baseline [3] method, and expert and suboptimal demonstrations in Section C of the Appendix of our revised paper to support this claim. It shows that our method’s performance closely aligns with that of the expert policy under all environments. On the contrary, the baseline [3] method rated high in Limited-Walker and Blocked-Ant environments while failing to achieve expert-level performance in the Obstacle-HalfCheetah environment.
>
> &nbsp;*7. 'In the MuJoCo experiments, is the reward presented in Table 2 the feasible reward? I.e. are rewards truncated after a constraint has been violated? It seems that that would be the more inveresting metric to report, I would recommend the authors report that metric, and if they already do so make it clear it is that metric.'*
>
> **Response.** Yes, the reward values reported in Table 2 represent the feasible reward, which is the cumulative reward obtained before any constraint violation occurs. This metric is consistent with the approach used in [3]. This clarification has been added to Table 2 in the revised version of the paper.
>
> **References**
>
> [1] Janner, Michael, et al. "Planning with Diffusion for Flexible Behavior Synthesis." International Conference on Machine Learning. PMLR, 2022.
>
> [2] Liu, G., Xu, S., Liu, S., Gaurav, A., Subramanian, S. G., \& Poupart, P. (2024). A Comprehensive Survey on Inverse Constrained Reinforcement Learning: Definitions, Progress and Challenges.
>
> [3] Guorui Quan, Zhiqiang Xu, Guiliang Liu (2024). Learning Constraints from Offline Demonstrations via Superior Distribution Correction Estimation. In Forty-first International Conference on Machine Learning.

---

> > ### Comment · Reviewer_56R9 · 2024-11-27
> >
> > I apologize for the tardiness of my response.
> >
> > The authors have satisfactorily answered my questions.
> >
> > It seems that the ICLR system does not allow for a score between 6 and 8, so I'll stick with my original score.

---

> > > ### Author Response · Authors · 2024-11-28
> > > **Author response to Reviewer 56R9 - Part 3**
> > >
> > > We are delighted by your positive assessment of our work and are most grateful for your thoughtful and thorough review of our manuscript.Your insightful comments have been invaluable in improving the clarity and precision of our work. We appreciate the significant time and effort dedicated to this process and welcome the opportunity to address your suggestions. Thank you very much!

---

### Official Review · Reviewer_zyAA · 2024-11-03

**Soundness:** 3
**Presentation:** 2
**Contribution:** 3
**Rating:** 6
**Confidence:** 3

**Summary:**

The paper tackles the safe reinforcement learning problem using a diffusion model and guidance to train a set of feasibility functions. Unlike traditional inverse constraint learning, which is difficult to verify whether a candidate constraint is feasible and returns a single constraint, the paper's algorithm rapidly recovers a diverse set of constraints once the diffusion model is trained on expert data. The paper's algorithm outperforms baselines on constrained mazes and Mujoco experiments regarding performance and sample efficiency.

**Strengths:**

- The idea of amortizing the ICL loop cost by pre-training a diffusion model is interesting.
- The paper provides convincing empirical results that show the superiority of their method compared to the baselines of their experiments, both for reward and cost. It also investigates how reliable feasibility functions are on expert non and non-expert data.
- While they have not directly demonstrated the advantages of having multiple constraint candidates returned by the algorithm (aside from possibly making search more efficient), this seems like a practical feature to have for real world use cases.

**Weaknesses:**

- While the authors list computational concerns as one of the advantages Ex-ICL has over ICL, they do not conclusively show Ex-ICL's computational advantage. Figure 6 shows that Ex-ICL is more sample efficient in constraint inference, but a true test of computational efficiency should also take into account diffusion model training time.
- The experiments on maze and Mujoco are comprehensive but are fairly simple. For example, the baseline paper [1] includes a more realistic experiment on traffic scenarios.
- There's not enough detail in the main paper or appendix on methodology (how is \phi parameterized?)

[1] Guorui Quan, Zhiqiang Xu, & Guiliang Liu (2024). Learning Constraints from Offline Demonstrations via Superior Distribution Correction Estimation. In Forty-first International Conference on Machine Learning.

**Questions:**

- How are you selecting the constraint out of the constraint pool discovered by Ex-ICL for the experiment section?
- Why does Figure 4's Ex-ICL figure have so much larger variance for bad trajectory cost value than other methods?
- How sensitive are the results to exploration coefficient \delta and exploration round m? Also, would it be instructive to showcase model performance for Ex-ICL that only searches over a single \delta?

---

> ### Author Response · Authors · 2024-11-22
> **Author response to Reviewer  zyAA - Part 1**
>
> Dear Reviewer, we greatly appreciate your constructive comments. We have seriously considered your suggestions, and hopefully, the following response can address your concerns:
>
> &nbsp;*1. While the authors list computational concerns as one of the advantages Ex-ICL has over ICL, they do not conclusively show Ex-ICL's computational advantage. Figure 6 shows that Ex-ICL is more sample efficient in constraint inference, but a true test of computational efficiency should also take into account diffusion model training time.*
>
> **Response.** We thank you for this important question. In this work, we prioritize the data-efficiency over computational resource consumption as it's a major concern of Offline Reinforcement Learning, which has been demonstrated in [2]. Such prioritization is due to that dataset collection other than model training is more time- and resources-costly in the offline learning pipeline. As shown in Figure 6, our method outperforms the previous Offline Inverse Constraint Reinforcement Learning methods in terms of this crucial consideration.
>
> As for diffusion model training, its data consumption has been taken into account. As supporting evidence, we refer you to the initial part of the Ex-ICL plot in the revised Figure 6 (within the orange ellipse). The consistent zero performance in this circled region indicates the training phase of the diffusion model and reward model.
>
> &nbsp;*2. 'The experiments on maze and Mujoco are comprehensive but are fairly simple. For example, the baseline paper [1] includes a more realistic experiment on traffic scenarios.'*
>
> **Response.** We appreciate you bringing this to our attention. To address this concern, we have conducted an additional experiment under the CommonRoad-RL environment with a velocity $<40$ constraint, following the experimental setup described in [1]. Using the same HighD expert dataset and suboptimal dataset for training as [1] ensures a fair comparison.
>
> As shown in section B.3 of the Appendix in the revised manuscript, our method demonstrated superior performance in terms of cost reduction while achieving comparable reward and Area Under Curve (AUC) performance to [1]. This result shows the potential of our method for realistic autonomous driving tasks.
>
> &nbsp;*3. 'There's not enough detail in the main paper or appendix on methodology (how is $\phi$ parameterized?)'*
>
> **Response.** Thanks for raising this important concern. We design a cost value model to model the feasibility $\phi$ in trajectory level, mirroring the reward value model architecture in the official implementation of [2], which employs a U-Net architecture. Given a horizon $H$-length noisy trajectory $\tau^i$ as input, the model predicts the feasibility $\phi_{\omega}(s_t, a_t)\in[0,1]$ for each state-action pair $(s_t, a_t)$ within the trajectory, generating a feasibility vector of length $H$. Based on these $\phi_{\omega}(s_t, a_t)$s, we compute the cost value $V_c = \sum_{t=0}^{H} \gamma^t c_{\omega}(s^i_t, a^i_t, i) = \sum_{t=0}^{H} \gamma^t (-\log \phi_{\omega}(s^i_t, a^i_t, i))$ for guiding the diffusion verifier. Note that the diffusion timestep, $i$, is provided as an explicit input to the network and embedded via an MLP to capture the timestep of the denoising process. Details of the cost value model architecture and hyperparameters are now included in Section A.4 and Table 3 of the Appendix in the revised manuscript.
>
> &nbsp;*4. 'How are you selecting the constraint out of the constraint pool discovered by Ex-ICL for the experiment section?'*
>
> **Response.** Thank you for raising this important point.
>
> Our experiments can be split into 3 parts: "5.1 Control Performance", "5.2 Exploratory Performance", and "5.3 Learning Efficiency".
>
> 1) For control performance (section 5.1), we selected the constraints discovered under the largest $\delta$. Since $\delta$ controls the level of regularization on sparsity, our setting is to align with the previous setting of ICRL solvers that favor the sparsity of constraints, such as Section 5.1 of [1], thereby providing a fair comparison with previous works.
>
> 2) For exploratory performance (Section 5.2), all the constraints stored in the constraint pool are included in the evaluation, with which we illustrated the diversity of learned constraints in Figure 4, 7, 8, and 9.
>
> 3) In terms of learning efficiency (Section 5.3), we illustrate the sample complexity in discovering the first valid constraint with EX-ICL. This is to offer a fair comparison with previous work that study only one constraint.
>
> These clarifications have been incorporated into the revised manuscript.

---

> ### Author Response · Authors · 2024-11-22
> **Author response to Reviewer zyAA - Part 2**
>
> &nbsp;*5. 'Why does Figure 4's Ex-ICL figure have so much larger variance for bad trajectory cost value than other methods?'*
>
> **Response.** We appreciate you raising this important point. In principle, to recover the constraint, ICRL algorithms increase the cost values of bad trajectories to be above the threshold $\epsilon$. On the other hand, for the cost values of expert trajectories, ICRL algorithms must guarantee their values to be smaller than  $\epsilon$. So the scale of costs for bad trajectories is much larger than those of expert trajectories, which causes significantly higher variance for bad trajectory cost value. By implementing strategic exploration, our EX-ICL exploration strategy successfully identifies this diverse set of feasible cost models, causing the variance of predicted cost values to be higher. We have revised the paper accordingly.
>
> &nbsp;*6. 'How sensitive are the results to exploration coefficient $\delta$ and exploration round m? Also, would it be instructive to showcase model performance for Ex-ICL that only searches over a single $\delta$?'*
>
> **Response.** We appreciate your insightful question.To assess the sensitivity to $\delta$, we conducted experiments across a range of exploration coefficients, the results of which are presented in Figures 4, 7, and 8. These figures illustrate the predicted cost values for models trained using different values of $\delta$ under different environments. Specifically, the initial model was trained with the smallest $\delta_0$. Five additional models were then trained using linearly increasing $\delta$ values: $\delta_i = \delta_0 + (\delta_5 - \delta_0) \times \frac{i}{5}$.
>
> To investigate the impact of the exploration rounds $m$, we have included a supplementary Figure 9 in Section B.1 of the revised paper. This figure shows the trend in cost values as the number of training rounds increases from 0 to $M$ using the largest value of $\delta$. This supplementary figure supports our claim that the difference in cost values between the initial model and the trained models grows significantly with increasing training rounds.
>
> **References**
>
> [1] Guorui Quan, Zhiqiang Xu, \& Guiliang Liu (2024). Learning Constraints from Offline Demonstrations via Superior Distribution Correction Estimation. In Forty-first International Conference on Machine Learning.
>
> [2] Janner, Michael, et al. "Planning with Diffusion for Flexible Behavior Synthesis." International Conference on Machine Learning. PMLR, 2022.

---

> > ### Comment · Reviewer_zyAA · 2024-11-25
> >
> > Thank you for your comprehensive response and additional experiments. I have raised my score to 6. One more question, why is BC's cost in Table 5 much lower than Ex-ICL? Does this suggest that ICL methods are unnecessary to avoid constraint violation for this problem?

---

> > > ### Author Response · Authors · 2024-11-25
> > > **Author response to Reviewer zyAA - Part 3**
> > >
> > > We would like to express our sincere gratitude for the reviewer’s constructive criticism and the considerable time and effort dedicated to evaluating our manuscript. Their insightful comments have proven invaluable in improving the clarity and overall quality of our work. Thanks a lot!
> > >
> > > As for the insightful concern the reviewer raised:
> > >
> > > &nbsp;*8. 'why is BC's cost in Table 5 much lower than Ex-ICL? Does this suggest that ICL methods are unnecessary to avoid constraint violation for this problem?'*
> > >
> > > **Response.** We appreciate the reviewer's important observation. In the meantime, another critical observation  is that BC yields a low cumulative reward, indicating a failure to meet the navigation objective within the designated time. This result suggests that there is no incentive for the BC agent to prioritize acceleration for reducing travel time, which consequently avoids violations of the velocity constraint. Essentially, BC learns a conservative or "dummy" policy that satisfies the constraint but is far from optimal. In contrast, our proposed method’s comparatively high reward indicates successful navigation. This necessitates a learning strategy that incorporates acceleration, increasing the likelihood of constraint violations and thus resulting in a higher cumulative cost > 0 rate. Importantly, the cost rate of our method remains lower than the ICSDICE [1] baseline, suggesting superior constraint modeling while maintaining a comparable reward performance.

---

> > > > ### Comment · Reviewer_zyAA · 2024-11-26
> > > >
> > > > Thank you for your informative reply.

---

### Official Review · Reviewer_X5Jn · 2024-11-04

**Soundness:** 2
**Presentation:** 2
**Contribution:** 2
**Rating:** 6
**Confidence:** 3

**Summary:**

This paper proposes ExICL to tackle Inverse Constraint Learning problem, which aims to recover a diverse set of feasible constraints through an exploratory constraint update mechanism. The designed generative diffusion verifier utilizes the guided sampling strategy to verify the feasibility of explored constraints. This paper also aims to guarantee the robustness of feasible constraints discovery by accurately estimating the cost of noisy trajectory.

**Strengths:**

1. Introduction clearly states the current issues in Inverse Constraint Learning and the related works section is complete.
2. The experiments are comprehensive, demonstrating the effectiveness of the proposed approach.

**Weaknesses:**

1. The contributions claimed in this paper are not apparent to me. Contents in 4.1 is quite close to what has been proposed in [1], and the non-convex objective theorem is inherited from [2]; the ambiguity of how things are defined in section 4.2, 4.3 impairs the significance of contributions again. There are many math notations are not defined or briefly mention. I will list each of them below in the question section. I found it confusing and hard to see how the idea works.
2. Again, theorem 4.1 seems related to some existing conclusion from Paternain's paper [2], and this theorem is critical as it supports the zero duality gap for non-convex objective. The theorem stated in this paper is not quite the same as what is shown in [2], as the constraints here are not constant but are functions, but constants in [2]. There is supposed to be a connection shown here to support the theorem or a direct proof. A typo follows the theorem in Equation (9): $\lambda\epsilon$ might be missing at the end in the exponential term.

[1]  Janner, Michael, et al. "Planning with Diffusion for Flexible Behavior Synthesis." International Conference on Machine Learning. PMLR, 2022.
[2] Paternain, Santiago, et al. "Constrained reinforcement learning has zero duality gap." Advances in Neural Information Processing Systems 32 (2019).

**Questions:**

1. My biggest confusion is about how the reward and cost are defined, respectively. Usually reward is defined as the negative cost if cost is positive, but in this paper, it seems not. Can you explicitly show how they are defined and how different they are?
2. In section 4.2, on line 286, how is $\phi_\omega(s_t^i, a_t^i, i)$ defined?
3. In section 4.3, can you explicitly give the expressions for dist$[1, \phi_\omega(s_t, a_t)$ and dist$[\tilde\phi_\omega(s_t, a_t), \phi_\omega(s_t, a_t)])$?
4. In algorithm 1, ``Updating $\lambda$ by minimizing the loss $\mathcal{L} = \lambda \mathbb{E}_{\hat\tau\sim \tilde{p}_M}[c(\tau) - \epsilon]$, why is no reward term involved here to update $\lambda$? Another question related to this in Table 2: there is a significant discrepancy between the magnitudes of the Reward and Cost. Could you provide some insight into this?

---

> ### Author Response · Authors · 2024-11-22
> **Author response to Reviewer X5Jn - Part 1**
>
> Dear Reviewer X5Jn,
>
> We sincerely value your time and effort in evaluating our work. We have prepared comprehensive responses and clarifications to address each point you raised. We hope these responses can resolve your concerns.
>
> &nbsp;*1. 'The contributions claimed in this paper are not apparent to me. Contents in 4.1 is quite close to what has been proposed in [1], and the non-convex objective theorem is inherited from [2]'*
>
> **Response.** Sorry for raising the confusion. We must first clarify that the core contribution of our research lies in the development of a novel method for learning a diverse set of feasible constraints from offline demonstration data. While certain constituent techniques may bear resemblance to prior work, their integration and application towards this specific objective distinguishes our approach.
>
> Although our generative diffusion verifier draws inspiration from [1], as noted in Section 4.1, several significant distinctions exist. Firstly, our utilization of the diffusion planner is fundamentally different. In our work, the diffusion planner serves the purpose of verifying learned constraints, rather than merely controlling the agent as in [1]. Secondly, our implementation incorporates a novel guidance mechanism utilizing an optimal probabilistic representation of the constrained policy model. This modification is substantiated by the findings presented in [2]. Critically, the subsequent stages of our methodology, namely the noise-robust constraint update and strategic exploration techniques, are entirely novel and unrelated to either [1] or [2]. These components are crucial to the success of our method in learning a diverse and robust set of constraints from offline data.
>
> &nbsp;*2. 'Again, theorem 4.1 seems related to some existing conclusion from Paternain's paper [2], and this theorem is critical as it supports the zero duality gap for non-convex objective. The theorem stated in this paper is not quite the same as what is shown in [2], as the constraints here are not constant but are functions, but constants in [2]. There is supposed to be a connection shown here to support the theorem or a direct proof. A typo follows the theorem in Equation (9): $\lambda\epsilon$ might be missing at the end in the exponential term.'*
>
> **Response.** We appreciate your careful reading of our manuscript and thank you for pointing out this important clarification. By "constants in [2]", we assume the reviewer indicates the cost $c(s_t, a_t)$ in [2] as a constant function given by the environment, providing the same output for identical inputs. In contrast, within our Inverse Constrained Reinforcement Learning (ICRL) framework, $c(s_t, a_t)$ is a time-varying function whose parameters are subject to updates.
>
> However, our ICRL training process is structured in two phases. The first phase focuses on constraint discovery, involving parameter updates for the cost model $c$. Subsequently, a Constrained Reinforcement Learning (CRL) phase is executed, updating the policy to comply with the discovered constraints. During this CRL phase, discovered constraints $c(s_t, a_t)$ behave as a constant function as its parameters are frozen. Therefore, Theorem 1 of [2], and its associated zero duality gap guarantee, remain valid within our framework. Algorithm 1 provides detailed information on this two-phase training process.
>
> Finally, we thank you for identifying the typographical error; this has been corrected in the revised manuscript.
>
> &nbsp; 3. *'My biggest confusion is about how the reward and cost are defined, respectively. Usually reward is defined as the negative cost if cost is positive, but in this paper, it seems not.'*
>
> **Response.** We thank you for your insightful question regarding the relationship between reward and cost in our work. It is important to clarify that the reward function in our framework is not simply defined as the negative cost. Since we address a Constrained Markov Decision Process (CMDP) problem, rewards and costs serve distinct roles in the policy optimization process. Specifically, agents are trained to maximize cumulative rewards. However, rather than directly minimizing cumulative costs, agents are trained to maintain cost values below a defined threshold, $\epsilon$. Section 3 provides a detailed explanation of the definitions of both reward and cost within our CMDP setting.
>
> If the reviewer is referring to [2], we wish to emphasize that the cost $c(s_t, a_t)$ in our work is functionally equivalent to $r_i(s_t, a_t)$ in [2]. In both cases, these functions are set to respect a cost/reward value threshold. While the cost value $\sum\limits^T_{t=0}\gamma^tc(s_t,a_t)$ is set to be smaller than $\epsilon$, the $i$-th constrained  $\sum\limits^T_{t=0}\gamma^tr_i(s_t,a_t)$ is set to be larger than $c_i$.

---

> ### Author Response · Authors · 2024-11-22
> **Author response to Reviewer X5Jn - Part 2**
>
> &nbsp;*4. 'In section 4.2, on line 286, how is $\phi(s^i_t, a^i_t, i)$ defined?'*
>
> **Response.** We appreciate your attention to detail. The notation $\phi(s^i_t, a^i_t, i)$ represents the feasibility of a state-action pair $(s^i_t, a^i_t)$ within a noisy trajectory $\tau^i$ generated during the $i$-th denoising step.
>
> To elaborate, the feasibility of a state-action pair $(s_t, a_t)$ at planning timestep $t$ is denoted by $\phi(s_t, a_t)$. However, this feasibility measure cannot be directly applied to guide the trajectory-level generation of diffusion verifier. As mirroring the approach in [1]'s official implementation, effective optimization of the diffusion verifier necessitates noise-robust reward and cost models capable of predicting values for noisy trajectories during the denoising process. Therefore, we explicitly incorporate the diffusion timestep $i$, resulting in the notation $\phi(s^i_t, a^i_t, i)$ to denote the feasibility of the state-action pair on the noisy trajectory $\tau^i$.
>
> This aspect has been clarified with additional detail regarding cost value model in Appendix A.4 of the revised manuscript.
>
> &nbsp;*5. 'In section 4.3, can you explicitly give the expressions for $\text{dist}(1, \phi)$ and $\text{dist}(\tilde{\phi}, \phi)$?'*
>
> **Response.** Thank you for your question. In our work, the distance function $\text{dist}$, is chosen to be the $l_1$-norm, as $\phi_{\omega}(s_t, a_t)$ is a scalar value. Consequently, $\text{dist} (1, \phi_{\omega}(s_t, a_t)) = |1 - \phi_{\omega}(s_t, a_t)|$ and $\text{dist} (\phi_{\omega}(s_t, a_t), \tilde{\phi}_ {\omega}(s_t, a_t)) = |\phi_{\omega}(s_t, a_t) - \tilde{\phi}_{\omega}(s_t, a_t)|$. This selection of the $l_1$-norm is now explicitly clarified in Section 4.3 of the revised manuscript.
>
> &nbsp;*6. 'In algorithm 1, ``Updating $\lambda$ by minimizing the loss $\lambda\mathbb{E}[c(\tau)-\epsilon]$, why is no reward term involved here to update?'*
>
> **Response.** We appreciate your question. This stems from the inherent properties of the standard Lagrangian approach to constrained optimization.
>
> As detailed in [2, 4], the dual problem for Constrained Reinforcement Learning (CRL) can be formulated as:
>
> $
> D^* = \arg\min_{\lambda} [V_r + \lambda(V_c - \epsilon)]
> $
>
> (see (14) in [2]). To update $\lambda$, we need to conduct the gradient descent for the above objective:
>
> $
> \lambda_{k+1} = \lambda_k - \eta \partial_{\lambda}d(\lambda)
> $
>
> based on equation (20) in [2],  and $\partial d(\lambda) = \partial_{\lambda}[V_r + \lambda(V_c - \epsilon)] = (V_c - \epsilon) = \mathbb{E}[c(\tau) - \epsilon]
> $.
>
> This leads to the loss function for $\lambda$ updates presented in our Algorithm 1: $\lambda \mathbb{E}[c(\tau) - \epsilon]$.
>
> As demonstrated in our derivation above, the reward term is not involved in the update of $\lambda$. The reward value $V_r$ is to be maximized in the original (PI) rather than to be constrained, thus it is decoupled from Lagrange multiplier $\lambda$ in the dual problem objective.
>
> &nbsp;*7. 'Another question related to this in Table 2: there is a significant discrepancy between the magnitudes of the Reward and Cost. Could you provide some insight into this?'*
>
> **Response.** Thanks for raising this concern. The discrepancy between the magnitudes of reward and cost in the experiments
> stems from the semantic difference and physical meaning distinctions between rewards and costs in the chosen environments.
>
> Specifically, in the MuJoCo locomotion environments detailed in Table 2, the cumulative feasible reward signifies the average distance traveled by the HalfCheetah, Walker, or Ant agent before either termination or violation of the constraints. Conversely, the cost represents the average number of timesteps during which the constraint was violated, measured within the 1000-timestep limit or until termination. Crucially, the environment assigns a cost of 1 for each timestep involving a constraint violation and a cost of 0 otherwise. This inherent distinction creates a significant discrepancy between reward and cost values. This reward and cost design is a standard practice within the Constrained Reinforcement Learning (CRL) and Inverse Constrained Reinforcement Learning (ICRL) literature, as noted in [3].
>
> **References**
>
> [1] Janner, Michael, et al. "Planning with Diffusion for Flexible Behavior Synthesis." International Conference on Machine Learning. PMLR, 2022.
>
> [2] Paternain, Santiago, et al. "Constrained reinforcement learning has zero duality gap." Advances in Neural Information Processing Systems 32 (2019).
>
> [3] Liu, G., Xu, S., Liu, S., Gaurav, A., Subramanian, S. G., \& Poupart, P. (2024). A Comprehensive Survey on Inverse Constrained Reinforcement Learning: Definitions, Progress and Challenges.
>
> [4] Boyd, S., \& Vandenberghe, L. (2004). Convex optimization.

---

> > ### Comment · Reviewer_X5Jn · 2024-11-25
> >
> > Thank you for the response and I will raise my score to 6 after reading this.

---

> ### Author Response · Authors · 2024-11-25
> **Author response to Reviewer X5Jn - Part 3**
>
> We are deeply grateful for the reviewer’s feedback and the significant time and effort invested in reviewing our manuscript. Their insightful comments have been invaluable in enhancing both the clarity of our work. Thank you very much!

---

### Author Response · Authors · 2024-11-22
**Summary of Updates and Global Responses**

Dear Reviewers, Area Chairs, and Program Chairs,

We sincerely appreciate your valuable feedback and insightful guidance. Your comments have been instrumental in significantly improving our work. In response, we have incorporated detailed clarifications, expanded explanations, additional experimental results, and improved figures into our revised manuscript (changes are highlighted in blue). A summary of the major updates is provided below:

1. More Details to Cost Value Model: Addressing the suggestions of Reviewers X5Jn and zyAA, we have clarified the U-Net-based architecture, the use of noisy trajectory inputs, the generation of pair-wise feasibility and trajectory cost value outputs, and the noise-robust nature of the model for guiding the denoising process.

2. Clarification on Constraints Selection: As requested by Reviewers zyAA and 56R9, we have detailed the $\delta$-based constraint selection process used in our experiments and further elaborated on the relationship between sparsity, the regularization parameter $\delta$, and its role in practical constraint selection.

3. Adding Experiment under Realistic Environment: Following the suggestion of Reviewer zyAA, we conducted additional experiments in a realistic autonomous driving environment to showcase our method's ability to handle complex real-world tasks.

4. More Experiments and Analysis on Cost Model's Performance: Addressing Reviewer zyAA’s suggestion, we have included additional experimental results illustrating the changing trend of cost model predictions as a function of the number of exploration rounds. In addition, to address the questions of Reviewer 56R9, we have provided a more in-depth analysis of the reasons behind the large variance observed in the exploratory cost models' predictions and the varying performance gap between our method and the baselines.

5. In-depth Explanation of Setting and Theory: Responding to Reviewer X5Jn's questions, we have provided a comprehensive explanation of the cost and reward definitions within our framework and detailed the duality optimization theory underlying our method's design.

---

### Comment · Area_Chair_oPsK · 2024-11-24
**Encouraging the reviewers to participate in discussion**

Hello, I encourage the reviewers to participate in discussion with the authors! Please recall that this phase of the discussion period ends on  Nov 26th, AoE.

To the authors eager to see author responses, I don't think there is anything to worry about yet. 3 days remain in the response period. Moreover it is currently the weekend (Saturday), and is thus understandable that the reviewers are not yet checking their OpenReview clients :)

---

### Meta-Review · Area_Chair_oPsK · 2024-12-17

**Metareview:**

This paper uses Diffusion models to develop new algorithms for in context reinforcement learning. While the sub-techniques are not novel, and there are no new theoretical contributions, authors generally appreciated the synthesis of existing ideas into a new application. Experiments showed promise, but reviewers felt that the domains could have been extensive, and had concerns that the proposed method did not uniformly outperform state of art. A more extensive set of experiments would be appreciated.

Overall, this paper is a nice application of existing ideas, and all three reviewers are okay with acceptance. However, no single reviewer feels incredibly strongly given the limitations mentioned above. Hence, I lean towards acceptance as a poster.

**Additional Comments On Reviewer Discussion:**

No reviewers were willing to champion the paper. One reviewer in particular said that they found the idea interesting, but would have liked to have since more extensive and compelling experimental results.

---

### Decision · Program_Chairs · 2025-01-22

Accept (Poster)